# Review of the Developments and Difficulties in Inorganic Solid-State Electrolytes

**DOI:** 10.3390/ma16062510

**Published:** 2023-03-21

**Authors:** Junlong Liu, Tao Wang, Jinjian Yu, Shuyang Li, Hong Ma, Xiaolong Liu

**Affiliations:** School of Materials, Sun Yat-sen University, Shenzhen 518107, China

**Keywords:** inorganic solid-state electrolytes, all-solid-state lithium-ion batteries, ion-transport mechanism, interface, air stability

## Abstract

All-solid-state lithium-ion batteries (ASSLIBs), with their exceptional attributes, have captured the attention of researchers. They offer a viable solution to the inherent flaws of traditional lithium-ion batteries. The crux of an ASSLB lies in its solid-state electrolyte (SSE) which shows higher stability and safety compared to liquid electrolyte. Additionally, it holds the promise of being compatible with Li metal anode, thereby realizing higher capacity. Inorganic SSEs have undergone tremendous developments in the last few decades; however, their practical applications still face difficulties such as the electrode–electrolyte interface, air stability, and so on. The structural composition of inorganic electrolytes is inherently linked to the advantages and difficulties they present. This article provides a comprehensive explanation of the development, structure, and Li-ion transport mechanism of representative inorganic SSEs. Moreover, corresponding difficulties such as interface issues and air stability as well as possible solutions are also discussed.

## 1. Introduction

Human civilization is heavily reliant on energy, but traditional fossil fuels such as coal, oil, and gas have a significant impact on the environment due to the release of carbon dioxide into the atmosphere, which is one of the primary contributors to global warming and poses a significant threat to the planet. Consequently, new energy sources such as wind and solar are gaining increasing importance as alternatives to traditional fossil fuels. However, these energy sources are variable in both space and time, necessitating the development of high-efficiency energy storage systems [1]. Among the various energy storage technologies available, Lithium-ion (Li-ion) batteries have emerged as one of the most promising options. The commercialization of Li-ion batteries by Sony in 1991 has led to their widespread adoption across numerous industries.

The use of unstable and flammable organic liquid electrolytes (OLEs) in conventional commercial Li-ion batteries poses a safety issue, and the assembly procedure is also rather complicated due to the possibility of liquid electrolyte leaks [2]. The best method for creating high-energy-density Li-ion batteries has been thought to be replacing the conventional anode with metallic Li. The capacity of today’s commercial lithium-ion batteries with liquid electrolyte is usually about 60~70 mAh·g^−1^, while many solid-state batteries using Li metal anode in the laboratory have an initial discharge capacity of more than 100 mAh·g^−1^. However, the application of Li metal anode has been hampered by the growth of Li dendrites, unstable Li/electrolyte interfaces, and Li pulverization during battery cycling, which limits the capacity of conventional Li-ion batteries [3,4]. In addition, Li dendrite is prone to form on the Li metal anode in OLEs, which can result in an internal short circuit of the battery. Assembling all-solid-state Li-ion batteries by replacing organic liquid electrolytes with non-flammable solid-state electrolytes (SSEs) is one option to totally address these safety issues [5]. 

The electrolyte is a critical component of an all-solid-state battery as it connects the two electrodes. Due to the stability between the electrolyte and the electrodes, which determines the capacity of the battery and restricts the types of electrode materials that can be employed, the electrolyte indirectly affects the battery’s capacity [6]. Solid electrolytes have distinct advantages over liquid electrolytes due to their stability and compatibility with a wider variety of electrode materials. This characteristic enables the use of electrodes that are not compatible with liquid electrolytes, allowing for the creation of batteries with higher capacities [7]. Additionally, solid electrolytes can serve as a separator, simplifying the battery’s structure and facilitating the assembly process [8].

Inorganic Li-ion conductors and polymer electrolytes are the two main categories of solid electrolytes. Polymer electrolytes are relatively versatile and can accommodate various battery shapes in addition to their improved safety and stability features [9]. However, polymer electrolytes are known to have weak heat stability and a limited electrochemical stability window, which can hinder their practical application in all-solid-state Li-ion batteries [10]. Inorganic Li-ion conductors have the advantage of non-flammability, wide operating temperature range as well as wide electrochemical stability window, which allows the all-solid-state Li-ion battery to operate over a broad voltage range [11,12].

In recent years, considerable research efforts have been dedicated to exploring the potential of solid-state electrolytes for various applications. Despite the promising prospects, the practicality of solid electrolytes remains a challenge, owing to a range of technical difficulties. High ionic conductivity (>10^−4^ S·cm^−1^ at room temperature) and electronic insulation are the fundamental requirements. The lower electronic conductivity prevents the accumulation of dead Li, which may form if Li ions gained electrons as they move through the solid-state electrolytes [11]. Reducing the contact resistance between the solid electrolyte and the electrode is another requirement for practical applications [13]. The practical application of solid electrolytes is severely constrained by the large interfacial resistance due to the high resistance and low stability at the interfaces leading to low Coulombic efficiencies, poor power performance, and short cycling lives. High electrical and chemical stability is necessary as electrolytes should be chemically stable enough to remain in contact with the electrode while cycling. Furthermore, it is critical to avoid the creation of by-products that affect the chemistry and kinetics of surfaces under atmospheric conditions. 

In this paper, the development process, structure, Li-ion transport mechanism, and stability of typical inorganic electrolytes are introduced. The difficulties posed by the electrolyte–electrode contact and the stability of electrolytes in the air are explored in detail, and associated progressions and solutions are introduced. Subsequently, the solid-state electrolytes and batteries mentioned in the article are summarized and compared with commercial lithium-ion batteries with liquid electrolytes. In the end, an outlook of inorganic solid-state electrolytes is proposed.

## 2. Solid-State Electrolytes

The two main categories of inorganic solid-state electrolytes are oxide and sulfide electrolytes. Although oxide electrolytes do not have particularly high ionic conductivity, they have good mechanical properties and stability to the atmosphere and electrode. Sulfide electrolytes with S^2−^ replacing O^2−^ of oxides show higher ionic conductivities at room temperature; however, due to their chemical reactivity with moisture, they are less stable in the air [14]. This section discusses oxide electrolytes such as garnet-type, perovskite-type, and NASICON-type structures, sulfide electrolytes such as thio-LISICON systems, and their stability issues.

### 2.1. Garnet-Type Electrolytes

#### 2.1.1. Historical Process, Structure, and Li-ion Diffusion Mechanisms

The chemical formula for natural garnet minerals is A_3_B_2_(SiO_4_)_3_, where A stands for eight-coordinated cations and B for six-coordinated cations. In 1968, a series of Li-containing garnets was first reported by Kasper with the formula of ${\mathrm{Ln}}_{3}^{3+}$M_2_${\mathrm{Li}}_{3}^{+}$ O_12_ (M = Te, W) [15], and Si ions are replaced by Li ions to form Li-O tetrahedrons, which is connected with the common vertices of the octahedron to form a three-dimensional framework. A new type of garnet structure, Li_5_La_3_M_2_O_12_ (M = Nb, Ta) was found by Mazza in 1988 [16]. In 2003, Thangadurai et al. were first to study the ${\mathrm{Li}}^{+}$ conduction in garnet Li_5_La_3_M_2_O_12_ (M = Nb, Ta) and reported a bulk conductivity in the range of 10^−6^ S·cm^−1^ at 25 °C [17], since then more and more garnet-type materials have been investigated as Li-Ion conductors. By replacing La with lower-valence cations (Ca, Sr, Ba [18]), Li6 garnet-type structures with increased conductivity can be created by increasing the Li concentration in the lattice. By substituting Zr with pentavalent cations in the lattice, Murugan et al. in 2007 found that the performance of LLZO (Li_7_La_3_Zr_2_O_12_) is particularly remarkable with the room temperature ionic conductivity of 3 × 10^−4^ S·cm^−1^ and lowest E_a_ = 0.3 eV as well as good thermal stability and chemical stability with metallic lithium [19]. Different composition of garnet-type Li ion conductors is shown in Figure 1, and there is a positive correlation between ionic conductivity and lithium ion content [20]. Although numerous studies synthesized LLZO, they did not obtain the desired outcomes. The conductivity of Li^+^ is affected by Li vaporization and a change in relative density during the sintering process [21]. To achieve high conductivity and stability of SSEs, controlling Li volatilization during sintering to produce samples with high Li content is desirable [22]. In 2009, Awaka et al. synthesized and analyzed tetragonal LLZO and found that this structure has only a conductivity of 1.63 × 10^−6^ S·cm^−1^ and E_a_ = 0.54 eV in the temperature range of 300–560 K [23].

Two crystal phases of LLZO exist: cubic phase (I_a-3d_) and tetragonal phase (I_41/acd_); and the tetragonal phase is the product of the lack of symmetry of the cubic phase. The two phases exhibit extremely distinct electrochemical properties. In tetragonal LLZO, the tetrahedral and octahedral sites are totally ordered by Li ions and vacancies, whereas the tetrahedral and octahedral sites of the cubic LLZO exhibit a complex Li-vacancy disordering. The crystal structure of tetragonal LLZO is shown in Figure 2a. There are three different sites of Li ions in tetragonal LLZO: Li(1) at the tetrahedral 8a site, Li(2) at the distorted octahedral 16f site, and Li(3) at distorted octahedral 32g site [23]. The migration path of lithium ions is along Li(2)-Li(3)-Li(1)-Li(3)-Li(2). All of the lithium ions migrate simultaneously to the following site as one of them goes down the path. The motion of Li ions in tetragonal LLZO is entirely collective because of the lack of available vacancies, which leads to low conductivity [24]. The crystal structure of cubic LLZO is shown in Figure 2b. The arrangement of Li in the cubic LLZO exhibits disorder for Li(1) at the tetrahedral 24d site and Li(2) at the distorted octahedral 48g/96h site [25]. Because certain Li(2) sites are partially occupied, cubic LLZO has high conductivity, and the diffusion path of lithium ions is Li(2)-Li(1)-Li(2) [26]. Due to the shorter distance between Li sites and the isotropic diffusion of Li ions, cubic LLZO shows much higher conductivity than tetragonal LLZO [27].

Because cubic LLZOs offer superior electrochemical characteristics, it is crucial to stabilize the cubic phase at ambient temperature. Researchers found that the cubic phase can be effectively stabilized by dopants. The grain boundaries are the bottleneck for ion transport, and the composition of the grain boundaries—which is primarily affected by dopants and Li segregation—strongly influences the grain boundary activation energy [28]. Ahn et al. studied the effect of the alumina crucible on the sintering of LLZO and proved that the incorporation of Al into the LLZO lattice can stabilize the cubic phase and improve its electrochemical performance [29]. It is suggested that both the lattice size and the integration of Al into the LLZO lattice, which increases the number of Li-ion vacancies, improve Li-ion conductivity. However, it is believed that Al^3+^ occupying the 24d sites will obstruct the migration of Li^+^ in its path [30]. Because the introduction of surplus Al into the bulk alters the preference of Al occupation at 96h sites over 24d sites, Kim et al. improved the phase stability of the LLZO, and ionic conductivity reached 3.84 × 10^−4^ S·cm^−1^ [31]. Ga doping exhibits a site preference similar to that of Al in the LLZO lattice, and the overall ionic conductivity of Ga-LLZO can be an order of magnitude higher than Al-LLZO [32]. Chen et al. simultaneously introduced Al and Ga into Li sites and prepared a series of AlxGa0.25-x-LLZO [33]. The room-temperature ionic conductivity of Ga0.25-LLZO reached 1.19 × 10^−3^ S·cm^−1^, and Ga doping not only promoted a stronger sinter ability with enhanced grain growth rate but also enabled a higher ionic conductivity. Another study explored the doping amount of Ga, the calcination temperature of Ga-LLZO primary powders, and the sintering conditions, Li_6.4_Ga_0.2_La_3_Zr_2_O_12_ sample with ionic conductivity of 1.25 × 10^−3^ S·cm^−1^ at 25 °C was obtained after calcining at a low temperature of 850 °C and sintering at 1100 °C for 320 min [34]. However, due to the formation of big grains in Ga-LLZO, which results in a loose grain boundary, Ga-LLZO is unstable against the Li anode [35]. Shin et al. co-doped Ta and Al into LLZO to achieve a similar result by having Al occupy the 96h site [30]. The multi-doped LLZO shows a much higher ionic conductivity of 6.14 × 10^−4^ S·cm^−1^ than that of the Al-LLZO. Nb doping can form contractile Nb-O bonds in LLZO, which can stabilize the cubic phase and speed up the sintering process [36,37]. Doping Ta makes more Li^+^ occupy Li2 sites and improves the conductivity between grain boundaries; however, it leads to low sinterability; Ga co-doping enhances the characteristics by providing good sinterability [38]. Wu et al. doped Rb to replace La and obtained Rb and Ga co-doping sample with a conductivity of 1.62 × 10^−3^ S·cm^−1^ at room temperature, which is the highest reported conductivity so far [39]. Another phase of the LLZO with the I_-43d_ space group has been proposed by Wagner et al. It results from the doping of large-radius ions such as Ga^3+^ or Fe^3+^ [40]. The decreased lattice symmetry in this structure leads to a significant improvement in Li^+^ diffusion [41]. With good chemical stability and high conductivity, the LLZO-based Li-ion conductor is a promising candidate for SSEs. However, due to the low stability with moisture and CO_2_ in the air as well as interface issue with electrodes, there is still a certain distance from commercialization [42].

#### 2.1.2. Stability toward Li Anode

LLZO-based electrolytes can coexist with Li anode in most situations. The intrinsic electrochemical stability window of LLZO reaches 0.05–2.91 V [43]. However, the form of the Li dendrites on the Li anode and the ineffective interfacial contact performance between metallic Li and LLZO hinder the application of LLZO [44]. The brittleness and rigidity of LLZO limit its close contact with the two electrodes, thus resulting in high interfacial resistance [45], and inhomogeneous deposition of lithium during Li plating/stripping processes could lead to dendrite growth [46]. The main goal of the current research is to find more effective solutions to these two problems.

The LLZO surface is reduced during the charge/discharge process together with Li^+^ implantation, resulting in a tetragonal LLZO interface that is stabilized at an extraordinarily thin thickness of around five unit cells as shown in Figure 3a. This interphase effectively stops subsequent interfacial reactions and keeps high ionic conductivity [47]. The reduction stability toward Li electrodes can be improved by adding more Li to the LLZO lattice [48]. However, LLZO has low wettability with the Li anode. The most common way to improve interface wettability is to use special materials to modify the electrolyte. Müller et al.’s magnetron sputtered In on LLZO to form a modified layer and improved cycle performance [49]. Jiang et al. introduced a thin layer of AlN between Li and LLZO, which reduces the interface impedance and improves the lithium-ion transport [50]. Fu et al. introduced Li-Al alloy to the interface, which changes the wettability of the garnet surface (from lithiophobic to lithiophilic) as Figure 3b shows [51]. Based on the ceramic LLZO and nonflammable trimethyl phosphate (TMP) gel, Li et al. developed a quasi-solid Janus electrolyte that has a broader electrochemical window and greater wettability [52].

Ga-doping LLZO is considered as one of the most promising solid-state electrolytes due to its high performance as Li-ion conductor. However, research has shown that Li_6.4_Ga_0.2_La_3_Zr_2_O_12_ can be reduced by Li at 25 °C [53]. As Ga leaches out of the lattice, LiGaO_2_ impurity forms at the Ga-LLZO/Li interface, resulting in a band-gap closure from >5 to <2 eV and a structural collapse from cubic to tetrahedral [54]. LiGaO_2_ impurity can react with Li to form Li_2_Ga alloy, which can cause the short-circuit of the battery. Li et al. added a trace of SiO_2_ during the synthesis of Ga-LLZO, which improve its stability toward Li metal [54]. The tiny amount of SiO_2_ captures Li from the LLZO matrix, creating Li vacancies that allow Ga to dissolve more easily in the garnet structure. As a result, the formation of the LiGaO_2_ impurity phase is eliminated, resulting in significantly improved stability. The Ga-LLZO-SiO_2_ electrolyte shows excellent circulation performance in the battery: (1) the Li/Ga-LLZO-SiO_2_(1 wt%)/Li half-cell operates stably for ∼500 h at a current density of 0.2 mA·cm^−2^ and then ∼1000 h at a current density of 0.3 mA·cm^−2^; (2) the LiFePO_4_/Ga-LLZO-SiO_2_(1 wt%)/Li full battery displays an initial discharge capacity of 155 mAh·g^−1^, a nearly 100% Coulombic efficiency, and a ∼99% capacity retention after the 20th discharge. Another study found that Ga-LLZO with small grain sizes is more stable to melting Li. Su et al. prepared fine-grained Ga-LLZO with grain size of 2–10 μm by a two-step sintering strategy [35]. The obtained Ga-LLZO has a high relative density of 97.3% and conductivity of 1.24 × 10^−3^·S·cm^−1^ at 27·°C. The symmetric cell with Li as electrodes shows a critical current density of 0.7 mA·cm^−2^ and a stable cycling of over 600 h at 0.4·mA·cm^−2^ at 27·°C. The Li/Ga-LLZO/LiFePO_4_ full cells deliver a reversible capacity of 150 mAh·g^−1^, showing negligible decay after 50 cycles.

Dendrite is prone to forming, especially during fast charging. LLZO with high density is more resistant to Li dendrites than porous LLZO which could be easily penetration by dendrites. Shen et al. found that using pure oxygen rather than air to assist in sintering produced LLZO with fewer pores, which had good stability and lengthy cycling even at a high current density of 0.4mA·cm^−2^ [55]. Botros et al. synthesized Al-LLZO through a novel route with nebulized spray pyrolysis and field-assisted sintering technology (FAST) to prepare dense ceramic electrolytes with a homogeneous elemental distribution [56]. Grissa et al. developed porous/dense/porous LLZO structures to increase the electrode/electrolyte contact surface and subsequently decrease the local current density at the interface to reduce the production of Li dendrites [57]. Liu et al. introduced a gel polymer electrolyte (GPE) into both sides of LLZO. The interface is improved by a flexible GPE buffer, and the rigid LLZO inhibits lithium dendrites. A Li|GPE@LLZO|LiCoO_2_ solid-state battery was assembled, and the battery showed a capacity retention of 82.6% after 100 cycles at 0.5 °C at room temperature [58]. He et al. used Cu-doped Li_3_Zn to guide uniform Li deposition by magnetron co-sputtering and an in situ alloying reaction on LLZO. As Figure 3c shows, the alloy layer transports the Li^+^ ions effectively and creates a connected intermediate layer, which prevents the growth of Li dendrite and provides good cycle performance (450 h at a current density of 0.8 mA·cm^−2^ without short circuit) [59].

#### 2.1.3. Stability toward Cathodes

LLZO-based electrolytes did not show excellent stability with cathodes such as ${\mathrm{LiCoO}}_{2}$ and NiCoMn (NCM) oxides. Side reactions frequently take place that could create an interlayer and reduce capacity. During the charging/discharging process, diffusion of cations occurs between LiCoO_2_ and the LLZO, and then an amorphous side phase with high impedance is formed, which results in capacity fading [60]. The NCM/LLZO interface where NCM and LLZO combine may experience Ni-La and Ni-Li exchange as well as Li diffusing into NCM to fill the Ni vacancy at high temperatures. After diffusion at the interface, the Li depletion layer causes the breakdown of LLZO and the creation of an interfacial layer composed of ${\mathrm{La}}_{2}{\mathrm{Zr}}_{2}O_{7}$ and LaNiO_3_ [61]. Although side reactions occur, LLZO is more stable to the cathode than most liquid electrolytes. Coating LLZO-based material on the cathode is regarded as a strategy to improve the cycle stability of batteries [62,63,64].

#### 2.1.4. Air Stability

LLZO-based electrolytes show low stability with moisture and CO_2_ in the air. Impurities such as LiOH and Li_2_CO_3_ could form in the air, especially with high humidity [42].
Li_7_La_3_Zr_2_O_12_ + xH_2_O → Li_7-x_H_3_La_3_Zr_2_O_12_ + LiOH
2LiOH + CO_2_ → Li_2_CO_3_ + H_2_O

LLZO is prone to Li^+^/H^+^ exchange and generates an H-LLZO interphase, which deteriorates the transport of Li^+^ across the LLZO/Li interface and leads to a decrease in conductivity [65]. The stability can be improved by doping strategies. Small-grain Al-LLZO reacts more slowly with air compared to big grains, which may contribute to the different distribution of Al and Li on the surface [66]. Ga-LLZO single crystal shows a short H^+^ diffusion length of 1 μm [67]. LLZO with Ga and Nb co-doping shows improved air stability because of inherent structural characteristics [68]. Although the Li_2_CO_3_ formed in the air can be effectively removed by polishing [69], improving air stability remains necessary for practical application.

### 2.2. Perovskite-Type Electrolytes

#### 2.2.1. Historical Process, Structure, and Li-ion Diffusion Mechanisms

The general formula for ideal perovskite-type materials is ABO_3_, where A and B represent six-fold and twelve-fold oxygen-coordinated cations, respectively. Figure 4a illustrates the ideal cubic perovskite with the space group P_m3m_. A-site can be occupied by cations with large ionic radii such as Na^+^, K^+^, Ca^2+^, Sr^2+^, Ba^2+^, La^3+^, etc., while B-site can be occupied by cations with small ionic radii such as Sc^3+^, In^3+^, Al^3+^, Sm^3+^, Ga^3+^, Ti^4+^, Zr^4+^, Hf^4+^, Sn^4+^, Ge^4+^, Nb^5+^, Ta^5+^, etc. Generally speaking, the perovskite-type Li-ion solid electrolytes are A-site-deficient materials, and the cubic and tetragonal perovskite crystal structures are the two main crystal structures [70]. In 1993, Inaguma et al. synthesized Li_0.34_La_0.51_TiO_0.294_, which exhibited a bulk conductivity exceeding 1 × 10^−3^ S·cm^−1^ at room temperature. However, due to the high grain boundary resistance, the overall conductivity of the obtained LLTO is only 2 × 10^−5^ S·cm^−1^ [71]. For LLTO, Li^+^ and La^3+^ occupy the A site and create 1/3-2x vacancy at the A site, while the B site was occupied withTi^4+^. Harada et al. prepared disordered cubic phase LLTO by heating to 1350 °C and quenching in liquid nitrogen, while tetragonal phase LLTO shows alternate Li-rich and La-rich layers in lattice [72]. Disordered cubic phase LLTO has a higher conductivity than tetragonal phase LLTO. Because it is challenging to prepare disordered cubic phase LLTO, tetragonal phase LLTO has received more attention.

For the tetragonal LLTO, La is unevenly distributed at La1 and La2 sites as Figure 4b shows, which leads to the doubling of the c-axis cell parameter and the tilting of TiO_6_ octahedra. Li-ions can contribute to conductivity since they are not bound to the hard lattice framework [70]. The La-rich layer and La-poor layer are known as La1 and La2 layers, respectively, which are oriented alternately along the C axis. La-rich layers are thought to be a barrier to Li conductivity. Li-ions can move from one vacancy to another through octahedral channels. There are two different pathways for Li-ions migration in the crystal structure: one migration pathway is along the ab plane, as shown in Figure 4c, and the other migration pathway is along the c-axis. ${\mathrm{Li}}^{+}$ typically migrate along the ab plane at low temperatures because the bottleneck size of the c-axis is significantly smaller than that of the ab plane [73]. The transport of Li ions in LLTO is anisotropic. The free volume for Li^+^ migration in the perovskite structure as well as the concentration of Li^+^ and vacancies on the A site have important effects on the ionic conductivity [74]. However, during the heat-treatment process of LLTO, a phase transform with La atoms moving to the original Li-rich layer is observed, which blocks the diffusion path of Li-ions and leads to a low ionic conductivity [75].

In order to improve the electrochemical performance of LLTO, there are many studies on the A-site and B-site doping of perovskite structures. Large rare-earth or alkaline-earth metal ions can be added to the A site to expand the bottleneck size, and the ionic conductivity increases with the size of the bottleneck [76]. For example, the incorporation of large-radius Sr into the lattice can effectively improve ionic conductivity [74]. For B-site substitution, the conductivity increases as the radius of the substituting ions decreases, which occurs because the decreased interatomic Ti–O bond distance strengthens the π_Ti-O_ bond, and the competing Li-O (σ_Li-O_) bond is weakened [77]. Since the grain boundary resistance is the main reason for the higher overall conductivity of LLTO, Ling et al. doped Ag into LLTO, which can assist the grain growth and decrease the grain boundary resistance, and thus improve the overall conductivity to 4.2×10^−5^ S·cm^−1^ [78]. In addition, the fabrication of LLTO electrolyte film is one of the methods to overcome the high grain boundary resistance [79].

Although the LLTO system has good electrical properties, Ti^4+^ in the system is easy to react with metallic Li, which limits its further application with Li anode. A perovskite-type system of (Li, Sr)(B, B′)O_3_ (B = Zr, Hf, Sn, etc., B’ = Nb, Ta, etc.) has been studied. Kimura et al. synthesized a series of Li_2x-y_Sr_1-x_Ta_y_Zr_1-y_O_3_ (LSTZ, x = 0.75 y) with various Ta contents (y = 0.60, 0.70, 0.75, 0.77, 0.8). For samples with y = 0.6 to 0.75, the maximum bulk and total conductivities reached 2.8 × 10^−4^ S·cm^−1^ and 2.0 × 10^−4^ S·cm^−1^ at 27 °C [80]. B. Huang et al. synthesized Li_3/8_Sr_7/16_Hf_1/4_Ta_3/4_O_3_ via a solid-state reaction method, which showed total conductivity of 3.8 × 10^−4^ S·cm^−1^, and electrical stability window was 1.4 V to at least 4.5 V vs. Li/Li^+^. [81]. Yu et al. synthesized Li_3/8_Sr_7/16_Hf_1/4_Nb_3/4_O_3_; however, they showed a lower conductivity of 2.0 × 10^−5^ S·cm^−1^ [82]. Amores et al. found a new series of lithium-rich double perovskites Li_1.5_Sr_1.5_MO_6_ (M = W^6+^, Te^6+^). Li_1.5_Sr_1.5_WO_6_ shows good potential as the anode, while Li_1.5_Sr_1.5_TeO_6_ is a good candidate for electrolyte with electrochemical stability up to 5 V and a low activation energy barrier (<0.2 eV) for microscopic lithium-ion diffusion [83].

#### 2.2.2. Stability toward Li Anode

Although LLTO is the most studied perovskite electrolyte, it does not show good stability towards Li anode. During cycling, Li dendrites prefer to nucleate at the voids in the LLTO pellets and grow vertically along the grain boundaries, which eventually lead to the cracking of LLTO as well as a short circuit of battery [84]. Even worse, LLTO has poor chemical stability with Li electrodes due to adverse reactions that occur during direct contact of LLTO with Li metal anodes:Ti^4+^ + Li → Ti^3+^ + Li^+^

The reaction leads to the decomposition of LLTO and influences on the diffusion of Li ions in electrolytes. In addition, LiO_2_ and La_2_O_3_ are observed in LLTO/Li anode interface. The LiO_2_ phase is formed as oxygen atoms contact with the Li-metal anode, followed by La being exposed to the interface and formation of the La_2_O_3_ phase [85]. Therefore, LLTO is difficult to apply alone in LIBs but often used in batteries as the ceramic filler of composite electrolytes, which plays a very good modifying effect. PEO is commonly used as an interface protection layer for perovskite electrolytes. Liu et al. prepared a flexible composite electrolyte comprised of a PEO-perovskite composite with a layer of PEO on either side. The design prevents direct contact between perovskite and lithium metal at the anode side, avoiding the undesired reaction between the two materials [86]. Jiang et al. used tape-casting to prepare LLTO electrolyte film with a thickness of 25 µm, and the total Li ionic conductivity of the film reached 2 × 10^−5^ S·cm^−1^ [87]. With PEO as protective layers, a full cell was assembled, showing an initial discharge capacity of 145 mAh·g^−1^ and a capacity retention ratio of 86.2% after 50 cycles. Yan et al. coated LLTO pellets with gel PEO-LiTFSI-SN (polyethylene oxide-lithium bis(trifluoromethanesulfonyl) imide-succinonitrile), which exhibit low interfacial resistance and improved chemical stability against Li metal without any sign of Li dendrite formation after 20 cycles [88]. Jia et al. prepared a composite electrolyte with LLTO particles coated by biodegradable polydopamine (PDA) layers and united with poly(vinylidene fluoride) PVDF. Which showed superior stability against Li-metal as well as outstanding flexibility and stretchability [89]. 

Ti-free perovskite-type electrolytes show much higher stability towards the Li anode. Xu et al. prepared a PEO/perovskite Li_3/8_Sr_7/16_Ta_3/4_Zr_1/4_O_3_ composite electrolyte. With a solid electrolyte interphase layer formed in situ between the metallic Li anode and the composite electrolyte, the formation and growth of Li dendrites were suppressed and the symmetric Li/composite electrolyte/Li battery exhibits an excellent cyclability at a high current density up to 0.6 mA·cm^−2^ [90].

#### 2.2.3. Air Stability

Similar to garnet-type solid electrolytes, LLTO could react with moisture and CO_2_ in the air. In the reaction, titanate and lithium carbonate are formed. The titanate could dissociate water on the grain surface and then exchange H^+^ for Li^+^ into the perovskite structure. The exchange happens more easily at higher temperatures [91]. LiOH would then form on the grain surface and react with CO_2_ in the air to generate Li_2_CO_3_ [92].

Recently, Li et al. reported a perovskite electrolyte composed of Li_0.38_Sr_0.44_Hf_0.3_O_2.95_F_0.05_ which is stable in moist air. The electrolyte has a Li-ion conductivity of 4.8 × 10^−4^ S·cm^−1^ at 25 °C and does not react with water having 3 ≤ pH ≤ 14 [93]. Although the mechanism is not clear, the outstanding stability of Li_0.38_Sr_0.44_Hf_0.3_O_2.95_F_0.05_ in moist air and aqueous solution contributes to keeping interface resistance at a low level of a Li-ion battery.

### 2.3. NASICON-Type Electrolytes

#### 2.3.1. Historical Process, Structure, and Li-Ion Diffusion Mechanisms

NASICON is the abbreviation of sodium super ion conductor. The NASICON-type ion conductor has a general chemical formula of AM_2_(PO_4_)_3_, where Na, Li, or K occupy site A, Ge, Zr, or Ti usually occupy site M [94]. The NASICON structure generally has a crystal phase of rhombohedral structure with space group R_-3c_, although monoclinic and orthorhombic phases have also been reported [95]. 

This structure has been studied as ion conductor since Goodenough et al. reported the synthesis and characterization of Na_1-x_Zr_2_Si_x_P_3-x_O_12_ (0 ≤ x ≤ 3) in 1976 [96]. In 1986, Subramanian et al. synthesized a series of LiM_2_(PO_4_)_3_ (B = Ti, Zr, and Hf) [97]. For NASICON-type LiM_2_(PO_4_)_3_, columns of MO_6_ octahedra are connected by PO_4_ tetrahedra. Li^+^ ions reside in two possible sites: the “Li1” site, which is 6-fold coordinated and located directly between two stacked MO_6_ units, and/or the “Li2” site, which lies in an 8-fold coordinated location between two columns of MO_6_ units. During long-range motion, the ions hop between these two sites as they traverse the crystal. The window between Li1 and Li2 sites, which is made of three O atoms bound to nearby M cations and is thought to be the bottleneck of Li-ion transport, controls the mobility of Li^+^ ions. Figure 5 shows the structure of a typical NASICON crystal with 3D conduction pathways. In the upper left, a close-up of the “bottleneck” area at the junction of three pathways is shown [98]. The size of the bottleneck is determined by the ions at the M site.

Casciola et al. studied the Li^+^ conduction in LiZr_2_(PO_4_)_3_ and reported an average conductivity of 7 × 10^−4^ S·cm^−1^ at 300 °C [99], which is much smaller than NaZr_2_(PO_4_)_3_. The reason is that the ion channel of [Zr_2_(PO_4_)_3_] skeleton is not suitable for Li^+^ migration. When the structure of the ionic conductor matches the size of migrating ions, the ionic conductor can achieve the maximum diffusion coefficient and the lowest activation energy [100]. Lu et al. reported Li_2.5_Sr_0.75_Zr_1.25_(PO_4_)_3_ with impressive proton conductivity of 0.178S·cm^−1^ at 550 °C [101]. Aono et al. studied NASICON-type LiM_2_(PO_4_)_3_ (M = Ge, Ti or Hf) and found that LiTi_2_(PO_4_)_3_ was the best NASICON-type electrolyte for Li-ion diffusion with the most suitable lattice size, the highest ionic conductivity, and the smallest activation energy for Li^+^ bulk migration [102]. Nevertheless, due to its poor sinterability, it still had a low conductivity. As Ti^4+^ in the structure exhibited poor stability especially after being in contact with metallic Li, another competitive candidate is LiGe_2_(PO_4_)_3_. The ionic conductivity of LiTi_2_(PO_4_)_3_ and LiGe_2_(PO_4_)_3_ solid electrolytes is significantly better than LiZr_2_(PO_4_)_3_. In addition to changing the bottleneck, another strategy to increase the conductivity of Li^+^ is the partial substitution of M^4+^ that increases the mobile lithium concentration and mobility [103]. Aono et al. used M^3+^(M = Al, Cr, Ga, Fe, Sc, In, Lu, Y, or La) to partially replace Ti^4+^ and prepared a series of Li_1+x_M_x_Ti_2-x_(PO_4_)_3_, and the sample with M = Al or Sc showed high ionic conductivity of 7 × 10^−4^ S·cm^−1^ [104]. Trivalent cation partial replacement of Ti^4+^ increases mobile lithium concentration and sinterability, thereby improving conductivity. Substitution of trivalent atoms not only increases the density of Li ions in the material but also induces additional interstitial migration with low activation energy, thereby affecting the mobility of Li ions [105]. The partial introduction of aluminum can stabilize the ionic conductivity higher than 10^−3^ S·cm^−1^ at room temperature [100], therefore, LATP and LAGP were widely studied in the past decades and have been regarded as representative of NASICON Li-ion conductors.

Li_1+x_Al_x_Ti_2-x_(PO_4_)_3_ (LATP) was reported by J. Fu in 1997, which exhibited an extremely high conductivity of 1.3 × 10^−3^ S·cm^−1^ at room temperature [106]. Hamao et al. fabricated a Li_1.3_Al_0.3_Ti_1.7_(PO_4_)_3_ electrolyte sheet by a cold sintering process (CSP), and the wettability of the electrolyte sheet was improved. The grain boundary was densified by the method, with the total conductivity of the prepared LATP sheet improved from 3 × 10^−6^ S·cm^−1^ to 3 × 10^−4^ S·cm^−1^ [107]. Xu et al. used the spark plasma sintering (SPS) technique to prepare a dense Li-ion conductor composed of nanostructured Li_1.4_Al_0.4_Ti_1.6_(PO_4_)_3_, which showed an ionic conductivity of 1.12 × 10^−3^ S·cm^−1^ and an activation energy of 0.25 eV at 25 °C [108]. 

The electrochemical window of the LATP-type material is limited by the low stability of Ti^4+^, which readily reduces to Ti^3+^ when in contact with the Li anode. Although Li_1+x_Al_x_Ti_2-x_(PO_4_)_3_ (LAGP) does not show as high conductivity as LATP, the high stability of LAGP makes it a popular material as an electrolyte. In 1997, Fu et al. reported Li_1.5_Al_0.5_Ge_1.5_(PO_4_)_3_ glass-ceramics with a conductivity over 10^−4^ S·cm^−1^ at room temperature [109]. Zallocco et al. synthesized Li_1.5_Al_0.5_Ge_1.5_(PO_4_)_3_ by glass sintering with concurrent crystallization [110]. Their sample showed ionic conductivity of 4.15 × 10^−4^ S·cm^−1^ and an electrochemical stability window around 3.5 V. Crystallization treatment is beneficial for obtaining glass-ceramics with higher electrical conductivity by controlling microstructures [111]. Zhu et al. prepared LAGP with the spark plasma sintering (SPS) technique, which could achieve LAGP pellets with high density, little voids and cracks, intimate grain-grain boundary, and high ionic conductivity of 3.29 × 10^−4^ S·cm^−1^ [112]. Xu et al. doped Li_2_O as a secondary phase in LAGP, which acted as a nucleating agent to significantly promote the crystallization of the as-prepared glasses during heat treatment, leading to an improvement in the connection between the glass-ceramic grains and resulting in a dense microstructure with a uniform grain size [113]. The LAGP-0.05LiO_2_ sample exhibits an ionic conductivity of 7.25 × 10^−4^ S·cm^−1^ and a stability window as high as 6 V. Nikodimos et al. enhanced the ionic conductivity of LGP by substituting 25% of Ge^4+^ ions in the LGP structure with Al^3+^ and Sc^3+^ ions, which leads to more Li+ in the vacant sites. The sample with the composition of Li_1.5_Al_0.33_Sc_0.17_Ge_1.5_(PO_4_)_3_ shows the highest bulk Li^+^ conductivity of 5.826 × 10^−3^ S·cm^−1^ with E_a_ of 0.279 eV [114].

#### 2.3.2. Stability toward Li Anode

LATP has low stability toward metallic Li because Ti^4+^ is easily converted to Ti^3+^, which causes LATP to degrade when in contact with the Li anode and lower ionic conductivity at the electrolyte–electrode interface. By replacing Ti^4+^ with Ge^4+^, LAGP is more stable and exhibits a wide electrochemical stability window of 0.85–7 V vs. Li/${\mathrm{Li}}^{+}$[115]. Although LAGP does not react with metallic Li at room temperature, it fails when it comes into contact with melting Li, severely restricting its applicability [116]. When contacting with melting Li at 330 °C, LAGP decomposes completely to form Li-based alloys, while LATP is partially decomposed without alloying [117]. The metallic lithium can penetrate the defect sites of the LATP bulk phase under elevated temperatures, which could lead to severe interfacial reactions [118]. Doping and creating a stable interlayer between the electrolyte and lithium metal are the two methods used to deal with the problem.

Mashekova et al. studied the effect of tetravalent and divalent cation dopants (Zr, Hf, Ca, Mg, Sr) of LATP on the Li-ion conduction and Ti reduction during interaction with lithium metal. The conductivity is not significantly impacted by a small ratio of doping, but its redox property is altered. Similar-sized tetravalent cations (Zr^4+^, Hf^4+^) appear to be suppressing the Ti^4+^ reduction, whereas large divalent cations (Ca^2+^, Sr^2+^) appear to be promoting it even at low concentrations. Doping of Mg^2+^ showed improvement in the Ti^4+^ reduction tolerance; however, it demonstrated a detrimental effect on the conductivity after contact with Li metal [119]. Stegmaier et al. conducted further research on the substitution of Mg^2+^ in LATP and found that the Mg^2+^ did not bleed heavily into the adjacent crystalline grain domains, which made it a suitable dopant for interfacial engineering. The small amount of Mg^2+^ ions that dope into the grain shows preferential substitution in the Ti/Al host framework, which preserves the functionality of the 3D Li network in LATP. Reduced Ti^4+^ content may, by polaron hopping, result in significantly lower residual electrical conductivity and shield the electrolyte from degradation [120]. Chen et al. added LiPO_2_F_2_ to modify the defect sites of the LATP pellet and impede the interfacial reactions between the LATP electrolyte and Li anode [118]. 

As the most commonly used method to improve battery cycle performance, interface engineering is also applicable. As shown in Figure 6a, Tolganbek et al. proposed the layer-by-layer polymer construction method for the ultra-thin interlayer of (PAA/PEO)_30_ on both sides of solid electrolyte pellets. The introduction of the protective layer prevented the formation of mixed conduction interphase and effectively decreased the interfacial impedance. The symmetric cell with Li metal electrodes and LATP-(PAA/PEO)_30_ electrolyte performed over 600 h at 0.1 mA·cm^−2^, and the modified LATP exhibited electrochemical stability up to 5 V [121]. Liu et al. reported LATP with Al_2_O_3_ coated by atomic layer deposition technique as Figure 6b exhibits. In comparison with bare LATP, the Al_2_O_3_-coated LATP exhibited a stable cycling behavior with smaller voltage hysteresis for 600 h, as well as small resistance. More importantly, the lithium penetration into the LATP bulk and Ti^4+^ reduction were significantly limited [122]. Huang et al. reported a MoS_2_ coating layer as an ASEI on LATP. The MoS_2_ layer not only effectively inhibits the decomposition of LATP, but also forms in situ a Li_2_S and Mo metal conversion layer during cycling, which can improve the interfacial charge transfer kinetics and decrease the charge transfer resistance [123]. The assembled Li/MCLATP/LFP cell shows excellent cycling performance over 300 cycles at 1 C. Cortes et al. deposited a thin (∼30 nm) Cr interlayer between the lithium anode and LAGP, which extend cycle life to over 1000 h at 0.1–0.2 mA·cm^−2^ from ∼30 h without protection [124]. The Cr interlayer promotes uniform interphase growth and delays fracture at moderate current densities thus extending the lifetime of cells. At the same time, the Cr layers promote reversible electrochemical conversion instead of Li deposition/stripping. Gao et al. reported a lithium-rich anti-perovskite Li_2_OHBr ionic conductor as a protective layer between LAGP and Li anode [125]. The Li_2_OHBr layer not only facilitates the migration of Li ions but also effectively avoids the reduction of LAGP by Li metal. 

#### 2.3.3. Stability toward Cathodes

During the charge and discharge process, ion exchange may happen between the electrolyte and cathode. Between LATP and LCO, Co-Ti exchange was found as Li^+^ ion moves with Co^3+^ to LATP, which led to high Li chemical potential around the LATP/LCO interfaces and the dynamical Li-ion depletion upon charging [126]. As a result of the reaction between LATP and LCO, a stable contact surface is created, which can prevent further reactions and promote capacity retention [127]. In addition, LFP reacts severely with LATP at low temperatures (T < 500 °C) and produces NASICON LiM_2_(PO_4_)_3_ (M = Fe and others), which remains as a thermodynamically stable phase at high temperatures in contact with the LATP [128]. When LATP is applied with NCM cathodes, it is observed that Ni^2+^ migrates from the nickel-rich cathodes to the NCM/LATP interface. Subsequently, Ni^2+^ reacts with Ti^4+^ in LATP, resulting in the decomposition of LATP. Wang et al. constructed a perovskite La_4_NiLiO_8_ buffer layer between NCM and LATP to reduce the content of Ni^2+^ on the surface, which enhanced the interfacial stability [129]. 

### 2.4. Thio-/LISICON System

#### 2.4.1. LISICON Structure Conductors

In 1978, Hong et al. were the first to synthesize a series of Li_16-2x_D_x_(TO_4_)_4_, where D = Mg^2+^ or Zn^2+^, T = Si^4+^ or Ge^4+^, and 0 < x < 4 [130]. Among these Li-ion conductors, Li_14_Zn(GeO_4_)_4_ exhibits ultrahigh conductivity 0.125 S·cm^−1^ at 300 °C, thus the structure was named Li superionic conductor (LISICON). The structure has the space group of P_nma_ and is similar to γ-Li_3_PO_4_ Each O^2−^ is bonded to four network cations (Li^+^, Zn^2+^, and Ge^4+^) to form a rigid three-dimensional network of Li_11_Zn(GeO_4_)_4_. The three remaining Li^+^ ions are weakly bonded with O^2−^ and are free to diffuse between the interstitial positions in the network. The average diameter of the bottlenecks between these positions is larger than twice the sum of the ionic radii of Li^+^ and O^2−^, satisfying the geometrical requirement for rapid Li^+^ transport. Ion transport in the structure is two-dimensional because the connected sites of mobile Li^+^ lie in the same plane. However, the conductivity of Li_14_Zn(GeO_4_)_4_ is only ~10^−6^ S·cm^−1^ at room temperature [131]. 

For the LISICON structure, high Li^+^ mobility appears to be favored by the introduction of interstitial Li^+^ ions in γ-tetrahedral structures. Hu et al. prepared Li_4-x_Si_1-x_P_x_O_4_ (0 < x < 1), the solid solutions formed between Li_4_SiO_4_ and Li_3_PO_4_. The samples with x = 0.5 and 0.6 were found to have the highest conductivity [132]. Si partially replaced P when Li_4_SiO_4_ was added to Li_3_PO_4_, and the number of Li^+^ ions occupying interstitial positions increased, thereby improving the ionic conductivity. Kuwano et al. synthetized the Li-ion conductor of the Li_4_GeO_4_-Li_3_VO_4_ system which shows higher ionic conductivity. The total conductivity of Li_3.6_Ge_0.6_V_0.4_O_4_ reached 4 × 10^−5^ S·cm^−1^ at 18 °C which is much higher than the previous LISICON structure [133]. High conductivity is attributable to the interstitial Li^+^ ions which are created during solid solution formation. In the γ-Li_3_VO_4_ cell, V^5+^ is partially substituted by Li^+^ and Ge^4+^ and additional Li^+^ ions occupy the interstice sites to form interstice Li^+^ ions in the Li_3+x_Ge_x_V_1-x_O_4_ crystal. The ionic conductivity is enhanced by the abundant Li^+^ ions in the interstice. Additionally, Li_3+x_Ge_x_V_1-x_O_4_ exhibits good thermal stability and is stable when in contact with the atmosphere of CO_2_. Deng et al. simulated and prepared a series of Si-substituted Li_4±x_Si_1-x_X_x_O_4_ (X = P^5+^, Al^3+^, or Ge^4+^). From low to high temperatures, three Li-ion diffusion mechanisms were identified: local oscillation, isolated hopping, and superionic motion. The substitution of P, Al, or Ge on Si sites caused polyanion mixing, which lower the temperature at which the transition to a superionic state and improved the room Li-ion conductivity by several orders of magnitude [134]. Zhao et al. prepared Li_3.75±y_(Ge_0.75_P_0.25_)_1-x_M_x_O_4_ (x = 0.1, M = Mg^2+^, B^3+^, Al^3+^, Ga^3+^, or V^5+^) and Li_3.75-y_(Ge_0.75_P_0.25_)_1-x_V_x_O_4_ (x = 0–0.5) to assess the effect of cation co-doping in LISICON structure, and Li_3.53_(Ge_0.75_P_0.25_)_0.7_V_0.3_O_4_ showed the highest ionic conductivity of 5.1 × 10^−5^ S·cm^−1^ at 25 °C. V^5+^co-doping lowers the energy barrier for Li^+^ diffusion in the lattice, which further increases ionic conductivity [135].

The effect of anionic substitution in the LISICON structures was also investigated. Song et al. reported a method for achieving high conductivity and great electrochemical stability by the substitution of Cl^−^ for O^2−^. As the radius of Cl^−^ is bigger than that of O^2−^, the substation of Cl^−^ for O^2−^ increases the lattice constants thus enlarging the bottleneck size, lowering the activation energy, and enhancing the ionic conductivity. However, the ionicity of Li-Cl is smaller than that of Li–O due to the lower electronegativity of Cl^−^. With increasing Cl concentration, the Li^+^ become tightly bonded to the Cl^−^, acting as an impediment to ion migration, increasing activation energy, and decreasing ionic conductivity. Therefore, appropriate Cl^−^ doping can significantly improve the properties of LISICON-type conductors. It was reported that Li_10.42_Si_1.5_P_1.5_C_l0.08_O_11.92_ and Li_10.42_Ge_1.5_P_1.5_C_l0.08_O_11.92_ show the highest ionic conductivities of 1.03 × 10^−5^ S·cm^−1^ and 3.7×10^−5^ S·cm^−1^ at 27 °C, respectively, which is an order of magnitude higher than Li_10.5_Ge_1.5_P_1.5_O_12_. With the introduction of Cl^−^, the electrochemical stability with lithium metal up to 9 V vs. Li^+^/Li, which is one of the widest electrochemical windows for solid electrolytes [136]. Fujimura et al. used first-principles calculations and machine-learning algorithms to predict the ionic conductivities of a great deal of LISICONs. It was predicted that Li_4_GeO_4_ will exhibit the highest ionic conductivities at 100 °C among the traditional LISICONs if it can be synthesized [137]. Overall, the insufficient ionic conductivity of LISICON-type solid electrolytes restricts their use in all-solid-state batteries and needs additional research. 

#### 2.4.2. Thio-LISICON Structure Conductors

The substitution of cations in the LISICON structure does not significantly improve the electrochemical performance, while the substitution of anions greatly changes the performance. The ionic conductivity can be significantly improved by replacing O^2−^ in LISICON-type solid electrolytes with S^2−^. Past research has shown that the size of the S^2−^ is near optimal for Li^+^ conduction in this structural framework [138]. 

Kanno et al. found six kinds of new thio-LISICON in the Li_2_S-GeS_2_, Li_2_S-GeS_2_-ZnS, Li_2_S-GeS_2_-Ga_2_S_3_ systems, which showed the highest conductivity of 6.5 × 10^−5^·S·cm^−1^ at room temperature [139]. The conductivity of thio-LISICON conductors could be significantly improved by the introduction of interstitial Li^+^. Liu et al. synthesized Li_2_ZrS_3_ through a solid-state reaction method and obtained Li_2_ZrS_3_ had ionic conductivity of 7.3 × 10^−6^ S·cm^−1^ at 30°C. With Zn^2+^ doping and partially substituting Zr^4+^, the Li_2.2_Zn_0.1_Zr_0.9_S_3_ exhibited much higher ionic conductivity of 1.2 × 10^−4^ S·cm^−1^ [140]. Masahiro prepared Li_3+5x_P_1-x_S_4_ (0<x<0.27) with different concentrations of Li^+^ in the structure. The interstitial Li created by partially replacing P with Li results in a substantial increase in the structure’s ionic conductivity and Li_3.325_P_0.935_S_4_ showed ionic conductivity of 1.5 × 10^−4^ S·cm^−1^ at 27 °C [141]. In 2002, Murayama et al. discovered Li_4+x_Si_1-x_Al_x_S_4_ and Li_4-x_Si_1-x_P_x_S_4_ in the Li_2_S-SiS_2_-Al_2_S_3_ and Li_2_S-SiS_2_-P_2_S_5_ system. The introduction of interstitial Li^+^ or Li^+^ vacancies in Li_4_SiS_4_ can improve ionic conductivity. Li_3.4_Si_0.4_P_0.6_S_4_ showed conductivity of 6.4 × 10^−4^ S·cm^−1^ and high electrochemical stability up to ~5 V vs Li at room temperature [142]. In the same year, Murayama et al. studied the structure of Li_4_GeS_4_ and observed the conductivity of Li_4-x_Ge_1-x_P_x_S_4_ and Li_4-x_Si_1-x_P_x_S_4_ solid solutions exceed 10^−3^ S·cm^−1^ and 10^−4^ S·cm^−1^, respectively [143]. 

Homma et al. reported three phase transitions (γ → β → α) of Li_3_PS_4_ during the increase of the experimental temperature. The main structure of Li_3_PS_4_ is composed of PS_4_ tetrahedral units, and the arrangement of PS_4_ tetrahedrons in the three phases is different, which affects the Li^+^ ion conduction. Among them, the structure of β-Li_3_PS_4_ is suitable for high ion conductivity and shows higher ionic conductivity. The structure of β-Li_3_PS_4_ is shown in Figure 7a, the PS_4_ tetrahedrons are segregated from one another and connected to the LiS_6_ octahedron via edge sharing. The PS_4_ tetrahedron’s apexes exhibit zig-zag arrangements. The LiS_6_ octahedrons are connected with each other via edge sharing, forming a one-dimensional (LiS_6_)_∞_ chain. Along the one-dimensional chain, the Li(1)S_4_ tetrahedrons are connected by corners, and an interstitial tetrahedral site exists between two Li(2)S_6_ octahedrons and one Li(3)S_4_ tetrahedron [144]. Liu et al. studied β-Li_3_PS_4_ with nanoporous structure, which exhibited a wide electrochemical window (5 V), superior chemical stability against lithium metal as well as an anomalous high ionic conductivity of 1.6 × 10^−4^ S·cm^−1^ at 25 °C, 3 orders higher than the intrinsic ionic conductivity of bulk β-Li_3_PS_4_ (8.93 × 10^−7^ S·cm^−1^). The nanoporous structure of β-Li_3_PS_4_ combines two important factors that improve ionic conductivity: (1) the stabilization of the high-conduction phase that develops at high temperatures and (2) the promotion of surface conduction by the high surface-to-bulk ratio of nanoporous β-Li3PS4 [145]. As the Li3PS4 system does not contain metal elements other than Li, it has high electrochemical stability [141]. 

In 2011, Kamaya et al. reported a new thio-LISICON-type Li_10_GeP_2_S_12_ (LGPS) with ionic conductivity of 1.2 × 10^−2^·S·cm^−1^ at room temperature, which was much higher than the solid-state Li conductors ever reported in recent decades and comparable to organic liquid electrolytes [146]. LGPS has the space group of P4_2_/nmc. As shown in Figure 7b, the unit cell has two tetrahedral sites: 4d and 2b sites. Ge and P ions occupy the 4d tetrahedral site, with occupancy values of 0.515(5) and 0.485(5), respectively. The 2b tetrahedral site is occupied by P with an occupancy parameter of 1.00(15). The unit cell contains three lithium sites: 16h, 4d, and 8f, with occupancy values of 0.691(5), 1.000(8), and 0.643(5), respectively. LiS_6_ octahedra and (Ge_0.5_P_0.5_)S_4_ tetrahedra connected by a shared edge to create one-dimensional (1D) chains, which form the framework of LPGS [146]. In LGPS, Li^+^ ions have two migration modes: along the tunnel in the c-axis and in the ab plane. The migration of Li^+^ along the c-axis is more favorable than that on the ab plane, which leads to weak anisotropy of Li^+^ migration of LGPS [147]. Although it is predicted that the diffusivity along the c-axis will be two orders of magnitude easier than that in the ab plane, there is still significant diffusion in the ab plane. At 300 K, the calculated conductivity in the ab plane is as high as ~1 × 10^−3^ S·cm^−1^, which is comparable to advanced solid electrolytes [148]. Liang et al. studied the two modes of Li^+^ ion diffusions in the LGPS by ^7^Li and ^31^P multiple solid-state NMR methods. Figure 7c depicts two distinct Li^+^ ion diffusion modes and the activation energy for diffusion in the tunnel along the c-axis is 0.16 eV, which is much lower than diffusion in the ab plane (E_a_ = 0.26eV) [149]. The strong Coulombic interaction amongst the mobile Li^+^ ions may be the origin of the collective migration process that results in the high conductivity along the c axis. As Figure 7d shows, when the Li^+^(1) hops to fill the adjacent vacancy, a large Coulombic repulsion against Li^+^(2) is generated, which prompts the next Li^+^ to jump to the latter position. In the synchronized action, the Li^+^ ions replace one another in a string-like fashion and this collective ionic migration exhibits a fairly low activation energy barrier [150]. Additionally, solid solutions of xLi_4_GeS_4_−yLi_3_PS_4_ with different x/y were studied (4/1 ≥ x/y ≥ 1/2). The room-temperature Li ion conductivity increases and subsequently declines as the Ge/P ratio increases, peaking at x/y = 1/1.2. The system with higher Ge content forms an ordered LGPS structure with slightly higher activation energy at room temperature [151].

Research on the application of LGPS has been put forward. Wang et al. synthesized Li_3.25_Ge_0.25_P_0.75_S_4_ through a low-temperature solution method, and the LGPS film exhibited a lithium-ion conductivity of 1.82 × 10^−4^ S·cm^−1^. Though the conductivity of the film is lower than bulk LGPS due to the film’s stoichiometry deviation and poor crystalline nature, it has a lithium-ion transference number of >0.999, which could be applicable in the thin film all-solid-state lithium-ion batteries [152]. Dawson et al. used nanoscale modeling techniques to simulate nanocrystalline LGPS systems with average grain sizes from 2 to 10 nm and predicted that the ionic conductivity of LGPS can be further improved by nanofication [153]. 

Generally speaking, with O^2−^ in LISICON-type solid electrolytes being replaced by S^2−^, thio-LISICON structure electrolytes show great improvement. Despite the necessity to address problems such as the high price of Ge and poor stability in damp air, LGPS is thought to have excellent application prospects since it exhibits an ionic conductivity comparable to organic liquid electrolytes. 

#### 2.4.3. Stability toward Li Anode

Sulfide electrolytes frequently do not have good stability with Li metal, and unstable interfaces can form. Take LGPS as an example, the intrinsic stability window of LGPS is very limited, from 1.7 to 2.1 V [43]. It was found that a mixed ionic electronic conducting interphase forms at the Li|LGPS interface, which grows fast. During the charging and discharging process, LGPS electrolytes decompose with the formation of Li_3_P, Li_2_S, and Li-Ge alloy, which increase the interfacial resistance [154]. More importantly, the formed SEI phase may continuously grow during cycling, which could destroy the cell eventually [138]. Thus, LGPS cannot be applied without protection against the lithium metal anode. The construction of an artificial SEI (ASEI) can effectively alleviate this problem. Kanno et al. studied the use of Li-Al alloy anode instead of Li anode [155]. Li-Al alloy and thio-LISICON (Li_3.25_Ge_0.25_P_0.75_S_4_) spontaneously form a breathing interface with the application of electric current, which effectively makes close contact at the electrolyte/electrode boundary and promotes fast charge transfer at the solid interface. For the Li anode, the SEI phase between the Li/thio-LISICON interface gradually grows during cycling and ultimately destroys the cell. Another study about Li-Al alloy anode also verified the excellent compatibility of Li-Al alloy and LGPS. Li_0.8_Al|LGPS|Li_0.8_Al could operate stably for more than 2500 h at 0.5 mA·cm^−2^ [156]. Gao et al. reported a nanocomposite consisting of organic elastomeric salts (LiO-(CH_2_O)_n_-Li) and inorganic nanoparticle salts (LiF, -NSO_2_-Li, Li_2_O) as interphase to protect LGPS from decomposition [157]. Figure 8a illustrates the interface between Li and LGPS with/without the protection of the nanocomposite layer. The nanocomposite is formed in situ on Li by electrochemically decomposing a liquid electrolyte and greatly improves the chemical stability at the interface. This nanocomposite layer not only has good stability and affinity for both the Li and LGPS but also provides fast ion conduction at the interface. A full cell using the decorated LGPS showed stable Li electrodeposition over 3000 h and a 200 cycles life. Zhang et al. prepared a manipulated LiH_2_PO_4_ protective layer on the Li anode to address intrinsic chemical stability problems of LGPS toward Li metal [158]. The stability of LGPS with Li metal significantly increased. The assembled LiCoO_2_|LGPS|Li ASSLiB showed a reversible discharge capacity of 131.1 mAh·g^−1^ at the initial cycle and 113.7 mAh·g^−1^ at the 500th cycle under 0.1 C with a retention of 86.7%.

Another crucial concern is the formation of lithium dendrites in solid-state batteries when the current density exceeds a threshold amount [159]. Introducing an interlayer between Li and the electrolyte is the most common approach to alleviate this problem. According to Su et al., the use of high external pressure with a Li/Graphite anode during testing of a solid-state battery using LGPS as the electrolyte resulted in a significant mechanical constraint on the level of the materials and increased rate performance of the battery. The solid-state battery could cycle at high current densities of up to 10 mA·cm^−2^ without lithium dendrite penetration or short circuit [160]. Additionally, the incorporation of Cl elements in sulfide electrolytes can inhibit the growth of Li dendrites to a certain extent [161].

#### 2.4.4. Stability toward Cathodes

Side reactions at the cathode during cycling are the main causes of electrolyte–cathode interface degradation. Zhang et al. studied the degradation of the interface between LGPS and LiCoO_2_. After cycling, side products such as sulfides are observed, accompanied by an increase in interfacial resistance and capacity fading. Three possible situations between LGPS and LiCoO_2_ during long-term cycling are proposed as Figure 8b shows. Situation I exhibits an ideal electrode–electrolyte interface, where Li ions are reversibly transferred between the cathode and SE, while electrons are transferred through the external circuit. In situation II, as the charge and discharge progress, the volumes of the electrolyte and electrode change, generating poor contact between the electrolyte and electrode, which leads to inhomogeneity of the local current density and nonuniform strain in materials. In situation III, the electrolyte decomposed with the formation of a Li depletion layer. The Li depletion layer can gradually grow, which further reduces the Li-ion mobility and causes more severe capacity fading [162]. Zuo et al. reported that the degradation kinetics of LGPS can be described by the Wagner-type model for diffusion-controlled reactions, indicating that the growth of the degradation layer generated at the electrode–electrolyte interface is constrained by electronic transport. Two oxidation mechanisms of LGPS are proposed: at medium potential (3.7 V vs. Li^+^/Li < E < 4.2 V vs. Li^+^/Li), LGPS decomposed with the formation of polysulfide species; at high potential (E ≥ 4.2 V vs. Li^+^/Li), the high potential triggers the structural instability and oxygen release at the cathode, which results in the formation of phosphate and sulfate species and more severe degradation [163]. In addition, the carbon additions in the composite cathode might encourage the electrochemical decomposition of the LGPS electrolyte, leading to undesired decomposition and high interfacial resistance [164].

#### 2.4.5. Air Stability

Although the ionic conductivity of thio-LISICON is significantly improved, thio-LISICON is not stable when exposed to air and moisture. The practical applications of thio-LISICON without modification are still limited. The P-S bond in the structure is unstable and can be easily replaced by other atoms to form a stable building block. As Figure 9 shows, the S(sulfur group) in the SSEs is readily hydrolyzed to form SH(mercapto group) and OH(hydroxyl group) in humid air, and then SH is further hydrolyzed to form OH and H_2_S [165]. Therefore, the S-based solid electrolytes are easy to hydrolyze in moist air and generate H_2_S gas, which leads to the collapse of structure and a decrease in conductivity [166]. The most common strategy to improve the air stability of thio-LISICON is partial substituting in structure based on hard and soft acids and bases (HSAB) theory. The HSAB theory proposes that soft acids/bases have high affinities for soft bases/acids, and hard acids/bases have high affinities for hard bases/acids, which manifests in the high stability of their products. Based on the HSAB theory, soft acids such as Sn^4+^, As^5+^, Sb^5+^ are inclined to bind tightly with the soft base S^2−^ and impede hydrolysis by H_2_O [167]. Similar to hard acid P^5+^ and hard base O^2−^, the introduction of O^2−^ into the structure shows significant improvement of air stability toward moisture in the air.

Zhang et al. reported the solid solutions of Li_4–x_Sb_x_Sn_1–x_S_4_ with 0 ≤ x ≤ 0.5 [168]. Li_3.8_Sb_0.2_Sn_0.8_S_4_ shows the highest ionic conductivity of 3.5 × 10^−4^ S·cm^−1^ among the system and high stability in moist air due to the high affinity of Sn-S and Sb-S. Liang et al. doped Sb^5+^ into LGPS as Sb^5+^ can form stronger covalent bonds with S^2−^ than P^5+^ [169]. The Li_10_Ge(P_1–x_Sb_x_)_2_S_12_ ionic conductors exhibit high air stability and have a high conductivity of 12.1~15.7 × 10^−3^ S·cm^−1^ even after being exposed to humid environment. 

The partial substitution of O^2−^ for S^2−^ in the structure can not only maintain the high conductivity of the sulfide prototype, but also improve the stability. Tsukasaki et al. replaced Li_2_S with Li_2_O in the 75Li_2_S·25P_2_S_5_ glass. The (75-x)Li_2_S·25P_2_S_5_·xLi_2_O could still maintain an ionic conductivity of higher than 10^−4^ S·cm^−1^ and better stability to air. For the samples with x ≥ 15, a good balance of thermal stability, wet resistance, and ionic conductivity can be expected [170]. Xu et al. synthesized the Li_9+δ_P_3+δ_′S_12–k_O_k_ series of solid solution phases and proposed that the solid solution range was 0 < k ≤ 3.6 based on the lattice parameter variation [171]. The sample with k = 0.9 shows the highest conductivity of 1.5 × 10^−3^ S·cm^−1^ as well as good stability in the air atmosphere. Liu et al. prepared a series of Li_10_Sn_0.95_P_2_S_11.9−x_O_x_ (0 ≤ x ≤ 1) by the solid-phase sintering method and Li_10_Sn_0.95_P_2_S_11.4_O_0.5_ shows high conductivity of 3.96 × 10^−3^ S·cm^−1^ with a negligible decrease after exposed in the air [172]. Gao et al. were first to prepare Sb^5+^ and O^2−^ substituted Li_10_SnP_2_S_12_ with high air stability due to the soft acid Sb^5+^ and hard base O^2−^ dual substitution [173]. The Li_10_SnP_1.84_Sb_0.16_S_11.6_O_0.4_ electrolyte displays a broader electrochemical window of 1.4–5.0 V vs. Li^+^/Li and ionic conductivity of 2.58 × 10^−3^ S·cm^−1^.

Although it is widely recognized that the water in the air can lead to severe decomposition of LGPS and release of H_2_S gas, resulting in a decrease in conductivity, Weng et al. reported a protective layer deliberately forming on the LGPS surface by controlling humidity [174]. In this research, LGPS is exposed in a humidity chamber with 45% humidity at 30 °C, and a protective decomposition layer of Li_4_P_2_S_6_, GeS_2_, and Li_2_HPO_3_ rapidly forms in dozens of seconds. The forming layer, which is ionically conductive and electronically insulating, can significantly prevent the severe interface reaction between LGPS and Li anode during electrochemical cycling. Both the symmetrical cell and full cell with LGPS exposed for 40 s exhibit good cycle performance. 

## 3. Summary

In this article, four typical inorganic solid-state electrolytes are introduced. Due to the difference in structure and composition, each solid electrolyte has its advantages and difficulties. The conductivities of mentioned electrolytes are summarized in Table 1.

Garnet-type electrolytes with prototype of LLZO can achieve high conductivity of ~10^−3^ S·cm^−1^ through doping strategy. Al-LLZO, Ga-LLZO, and Ta-LLZO exhibit high conductivity and have been extensively studied. Most garnets have relatively good chemical stability to Li; however, Ga-LLZO is prone to undesirable side reactions with Li. In addition, Garnet-type electrolyte is not stable in contact with moisture and CO_2_ in the ambient atmosphere. Therefore, the use of Garnet-type electrolytes in a battery needs modification to improve stability.

The application of perovskite-type electrolytes is limited due to low total conductivity led by high grain boundary resistance. The conductivity of perovskite electrolytes is only on the order of 10^−4^ or 10^−5^ S·cm^−1^. LLTO has poor stability towards Li and Ti-free perovskite-type electrolytes can solve this problem to some extent. Perovskite-type electrolyte exhibits high compatibility with commonly used cathode materials and can be used as coating layer on cathodes.

As the most representative NASICON-type electrolytes, LATP and LAGP show high ionic conductivity and moisture/air stability; however, they have low stability towards Li anode. The commercialization of LAGP is limited by the high cost of Ge-containing raw materials.

LISICON-type electrolytes exhibit good chemical and electrochemical stability but low conductivity of ~10^−5^ S·cm^−1^. Thio-LISICON-type electrolytes show much higher ionic conductivity with O^2−^ replaced by S^2−^. Among them, LGPS has the highest conductivity of ~10^−2^ at room temperature but low air stability due to its reaction with moisture. After partial substitution based on HSAB theory, the electrolyte still has a sufficiently high conductivity of ~10^−3^ with higher stability.

Commercial liquid electrolytes usually use DMC, EC, or PC as mixed solvents, LiPF6 as ion-conductive salt, and have the conductivity of the order of 10^−3^~10^−2^ S·cm^−1^. Some of the solid-state electrolytes show comparable ionic conductivity and are considered to be promising candidates for all-solid-state Li-ion battery.

For all-solid-state Li-ion batteries with Li as anode, the formation of dendrites is a problem that cannot be ignored. Solid-state electrolytes with high density can resist penetration by Li dendrites to a certain extent; however, the risk of short circuit persists after a long-time cycle. Therefore, it is necessary to make the current even distribution and regulate the deposition of Li by interface modification. The performances of mentioned cell with modified electrolytes are summarized in Table 2. 

The commercial lithium-ion secondary batteries produced by Panasonic have capacity of about 60~70 mAh·g^−1^. CALT recently reported a Qilin battery with an energy density of 255 Wh·kg^−1^, which can be converted to a capacity of 63.75 mAh·g^−1^ at 4 V. As can be seen from Table 2, many ASSLIBs show high capacity of over 100 mAh·g^−1^ with an operating voltage of about 4 V, which is much higher than commercial lithium-ion batteries with liquid electrolytes. However, the cycle perforsmance of some solid-state batteries is poor, which means that there is still a certain distance from commercialization standards for many batteries in the laboratory. Although some Li-ion batteries with solid-state electrolytes in the laboratory show good performance, it is challenging to commercialize them because of their high cost.

## 4. Conclusions and Outlook

This review concludes the research on four types of representative inorganic SSEs: garnet, perovskite, NASICON, and thio-/LISICON. The crystal structures of these SSEs, Li-ion transport mechanism, and ionic conductivity improvement techniques are all briefly discussed. Interfaces between solid-state electrolytes and anode/cathode, air stability of solid-state electrolytes, and associated improvement strategies are also addressed.

Compared to liquid electrolytes, SSEs have higher stability and safety as well as the prospect of being compatible with Li metal anode to achieve higher capacity. However, the rigid nature of solid-state electrolytes and poor contact of solid–solid interface lead to high interface resistance and non-uniform current distribution at the interface. Furthermore, the battery’s ability to retain high capacity and long-term processing has always been hampered by issues with the stability of the electrolyte and electrodes at their interface and stability in the air. Although substantial achievements have been made so far, there is still a long way to go before the successful industrialization and commercialization of all-solid-state lithium-ion batteries. Based on the aforementioned research, the following elements of future solid-state electrolytes need to be improved.

(1) Improving the ionic conductivity of solid-state electrolytes: Most solid electrolytes still lag far behind organic liquid electrolytes in ionic conductivity. On the one hand, the material composition, structure, and doping strategy of ionic conductors directly determine the ionic conductivity of the material. On the other hand, the preparation process affects the properties of the obtained product. It is important to optimize material composition and preparation process in order to improve ionic conductivity.

(2) Improving the stability of solid-state electrolytes toward the Li metal anode: Compatibility with high-capacity Li electrodes is one of the advantages of solid-state electrolytes, though the use of lithium metal electrodes is subject to certain limitations. The inclusion of an appropriately modified layer can minimize the growth of Li dendrites while also significantly enhancing the stability and contact of the interface. Additionally, the use of a lithium alloy anode can significantly lower the production of Li dendrites, and the stability of electrolytes can be increased by using the right doping and manufacturing techniques. 

(3) Improving the stability of solid-state electrolytes toward cathodes: The reactions at the interface between cathodes and electrolytes are very complex. At the interface, the cathodes and electrolytes break down, which might create a layer of depleted lithium and cause the interface to degrade. Exploring electrolytes with higher oxidation stability is one solution. In addition, constructing a suitable artificial solid-electrolyte interface layer also contributes to the improvement of stability.

(4) Improving the air stability of solid-state electrolytes: When electrolytes are exposed to air for a long period, they may react with the H_2_O and/or CO_2_ in the air. Sulfide electrolytes are particularly poorly stable to moisture in the air. Doping modification is the main method to improve the stability to the air, and attention should also be paid to the maintenance of electrolytes’ properties.

(5) Reducing costs and optimizing processes: Through unique synthesis techniques and interface alterations, solid-state electrolytes have been improved in numerous previous studies, making them more appropriate for all-solid-state battery operations over a lengthy period. However, some methods greatly increase the cost and complexity of production, which is not suitable for practical industrialization and commercialization.

Numerous types of electrolyte materials have been proposed as a result of the growing depth of the study on solid electrolytes. However, the intrinsic properties of these materials cannot fully meet all the requirements of solid-state batteries. The stability of the electrode–electrolyte interface is the most difficult issue to solve, and it will also be the subject of a future study. On the one hand, it is required to enhance the intrinsic qualities of electrolyte materials to make the procedure easier and the cost of modification lower. On the other hand, the practical application must investigate optimization techniques that can be used in commercial and industrial production.

## Figures and Tables

**Figure 1 materials-16-02510-f001:**
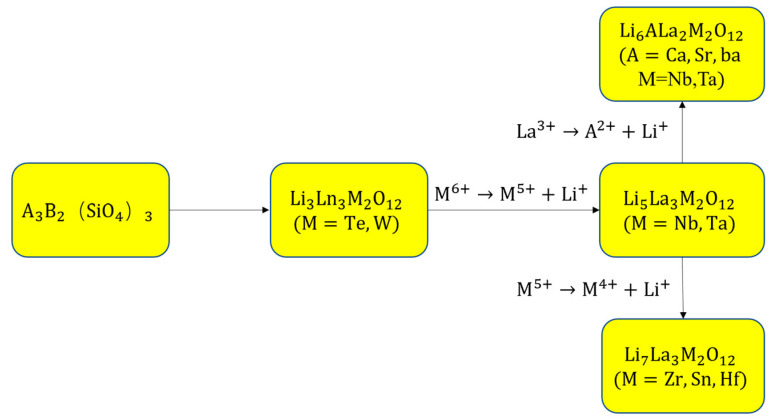
Different composition of garnet-type Li-ion conductors.

**Figure 2 materials-16-02510-f002:**
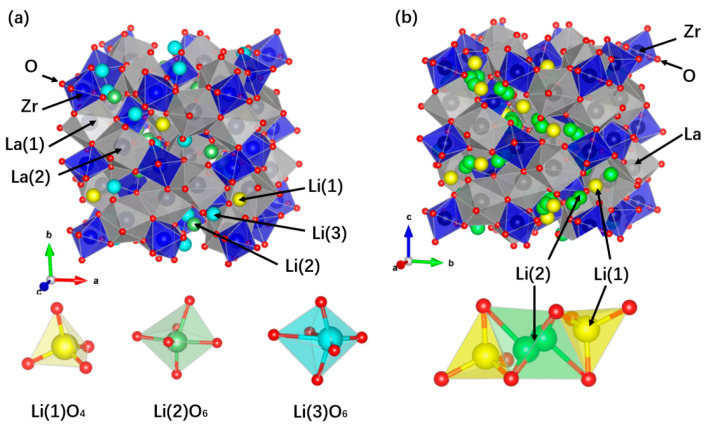
Crystal structure of (**a**) tetragonal LLZO and (**b**) cubic LLZO, and corresponding Li sites.

**Figure 3 materials-16-02510-f003:**
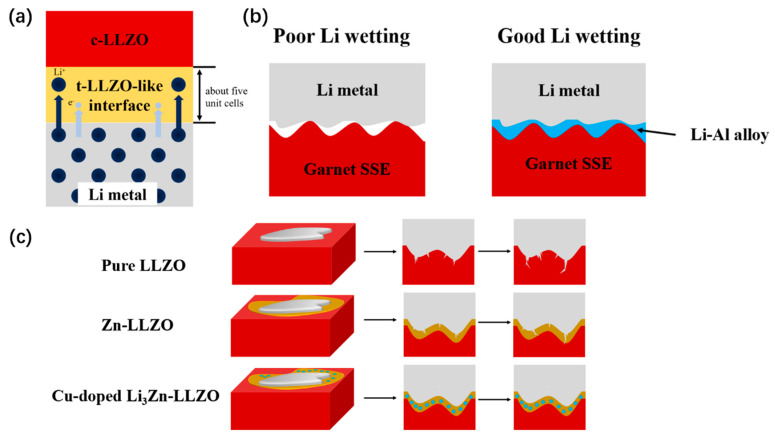
(**a**) Schematic of the interface behavior between c-LLZO and Li metal. A t-LLZO-like interphase is formed because c-LLZO is reduced and receives Li^+^ from Li metal. (**b**) Schematic of garnet SSE/Li interface with Li-Al alloy. The pristine garnet SSE has poor contact with Li. Al-coated garnet SSE exhibits good contact with Li due to the formation of Li-Al alloy. (**c**) Schematics of lithium deposition on the pure, Zn- and Cu-doped Li_3_Zn-LLZO’s interface. The Cu-doped Li_3_Zn interface prevents the growth of Li dendrite.

**Figure 4 materials-16-02510-f004:**
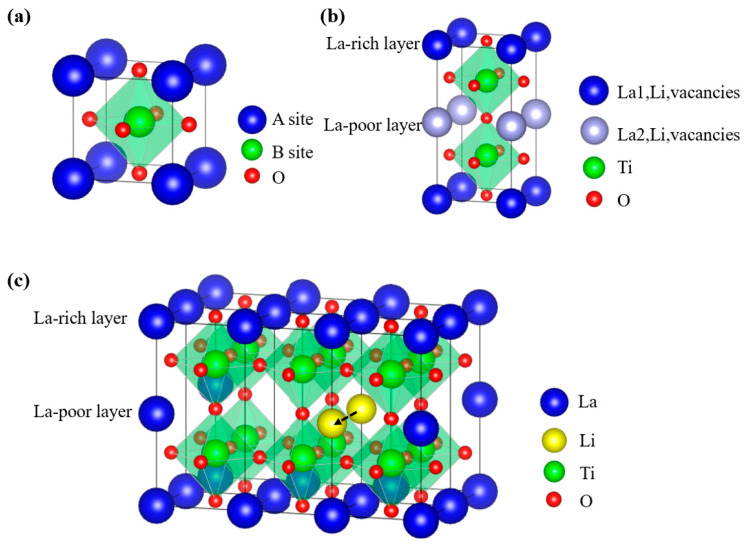
(**a**) ABO_3_ structure. (**b**) Crystal structure of LLTO with La-rich and La-poor regions. (**c**) Schematic diagram of the supercell of LLTO and Li pathway.

**Figure 5 materials-16-02510-f005:**
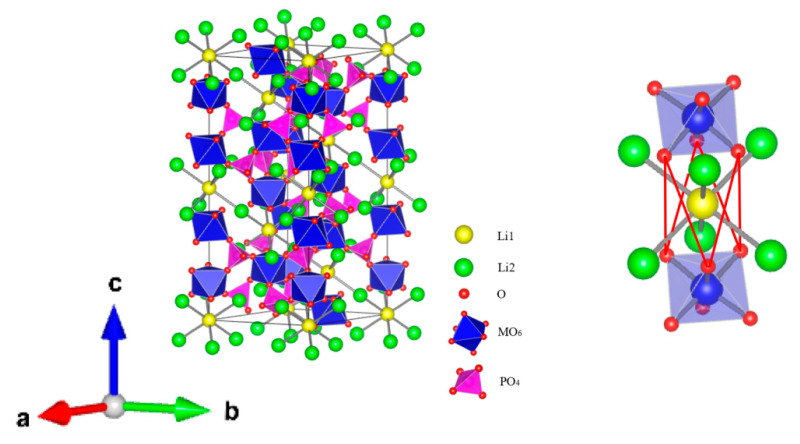
Crystal structure of NASICON-type LiM_2_(PO_4_)_3_. Li^+^ motion pathways are drawn between Li1 and Li2 sites. The structure on the right is a close-up view of the conduction bottleneck region and the constrictive windows between O atoms are traced in red.

**Figure 6 materials-16-02510-f006:**
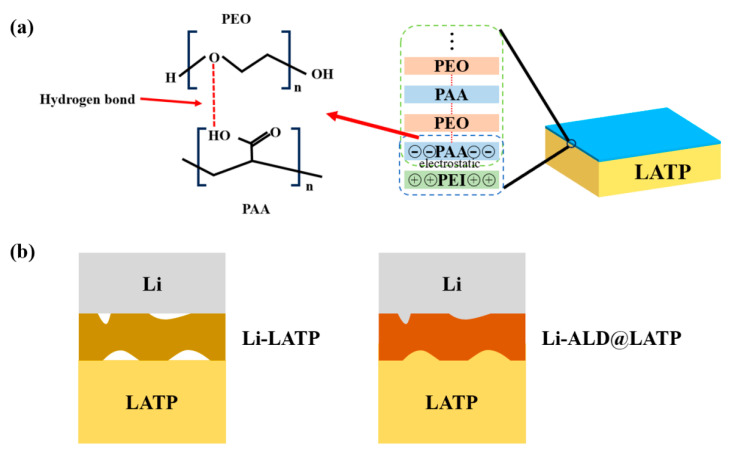
(**a**) Schematic of ${\left(\mathrm{PAA}/\mathrm{PEO}\right)}_{30}$ layer on the LATP. (**b**) Schematic of LATP/Li interface of LATP coated with Al_2_O_3_ by ALD.

**Figure 7 materials-16-02510-f007:**
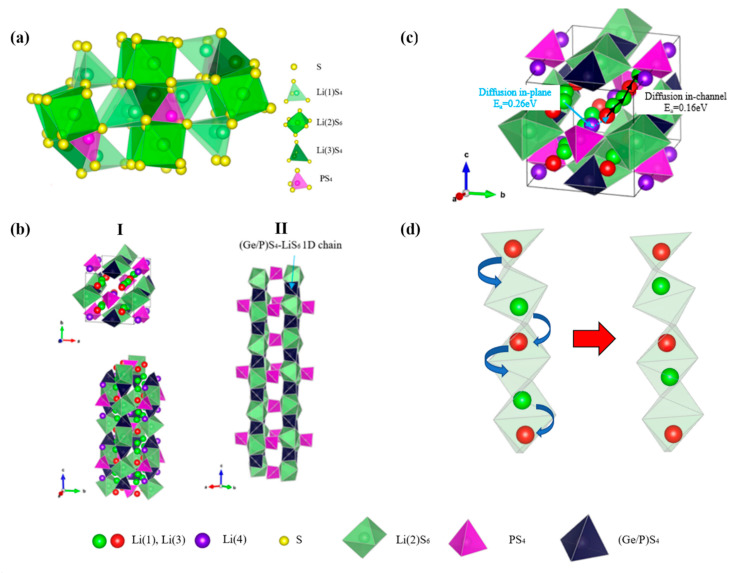
(**a**) Crystal structure of β-Li_3_PS_4_. (**b**) Crystal structure of Li_10_GeP_2_S_12_. (I) Structure of Li_10_GeP_2_S_12_ and lithium ions involved in ionic conduction. (II) Framework of Li_10_GeP_2_S_12_. (**c**) Schematic of two Li^+^ ion diffusion modes in Li_10_GeP_2_S_12_. (**d**) Schematic of the correlated hopping of Li ions diffusion in the channel.

**Figure 8 materials-16-02510-f008:**
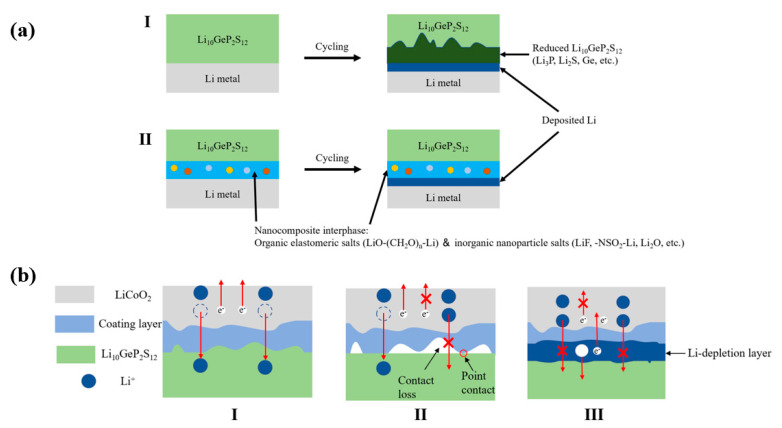
(**a**) Schematic of the Li/LGPS interface with Li salt-based organic-inorganic nanocomposite as an interfacial protective layer. (I) Li and LGPS show poor interfacial stability. LGPS is reduced by metallic Li, and some poorly ionic conductive products (Li_3_P, Li_2_S, Ge, etc.) are formed at the interface. (II) The stability of the Li/LGPS interface is improved by nanocomposite interphase consisting of organic elastomeric Li salts (LiO-(CH_2_O)_n_-Li) and inorganic nanoparticle salts (LiF, -NSO_2_-Li, Li_2_O). (**b**) Schematic of three possible situations occurring in the interface between LiCoO_2_ and LGPS during cycling. (I) Ideal electrode–electrolyte interface. (II) Poor contact between the electrolyte and electrode after cycling. (III) Li depletion layer forms between the electrolyte and electrode.

**Figure 9 materials-16-02510-f009:**
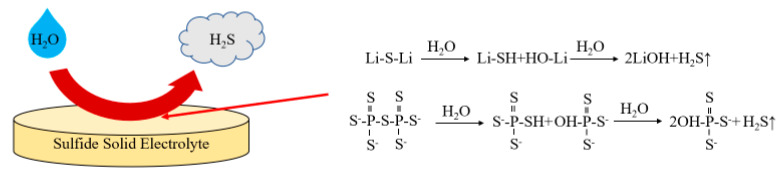
Instability of sulfide electrolytes in moist air and corresponding reactions.

**Table 1 materials-16-02510-t001:** Summary of ionic conductivity of solid electrolytes.

Electrolyte	Composition	Ionic Conductivity (S·cm^−1^)	Ref.
Garnet	Li_5_La_3_M_2_O_12_ (M = Nb, Ta)	10^−6^ S·cm^−1^ (25 °C)	[17]
	Li_7_La_3_Zr_2_O_12_	3 × 10^−4^ S·cm^−1^ (RT)	[19]
	Li_7_La_3_Zr_2_O_12_(tetragonal)	1.63 × 10^−6^ S·cm^−1^ (RT)	[23]
	Li_6.4_La_3_Zr_2_Al_0.2_O_12_	3.8 × 10^−4^ S·cm^−1^ (RT)	[31]
	Li_6.25_La_3_Zr_2_Ga_0.25_O_12_	1.19 × 10^−3^ S·cm^−1^ (RT)	[33]
	Li_6.4_Ga_0.2_La_3_Zr_2_O_12_	1.25 × 10^−3^ S·cm^−1^ (25 °C)	[34]
	Li_6.4_Ga_0.2_La_3_Zr_2_O_12_	1.24 × 10^−3^ S·cm^−1^ (27 °C)	[35]
	Li_6.2_Al_0.2_La_3_Zr_1.8_Ta_0.2_O_12_	6.14 × 10^−4^ S·cm^−1^ (RT)	[30]
	Li_6.6_La_3_Zr_1.6_Nb_0.4_O_12_	3.86 × 10 ^−4^ S·cm ^−1^ (RT)	[36]
	Li_6.4_Ga_0.133_La_3_Zr_1.8_Ta_0.2_O_12_	6.141 × 10^−4^ S cm^−1^ (RT)	[38]
	Li_6.20_Ga_0.30_La_2.95_Rb_0.05_Zr_2_O_12_	1.62 × 10^−3^ S·cm^−1^ (RT)	[39]
		4.56 × 10^−3^ S·cm^−1^ (60 °C)	[39]
Perovskite	Li_0.34_La_0.51_TiO_0.294_	2 × 10^−5^ S·cm^−1^ (RT)	[71]
	Li_0.5_La_0.5_TiO_3_/5 wt% Ag	4.2 × 10^−5^ S·cm^−1^ (RT)	[78]
	Li_0.3_Sr_0.65_Ta_0.6_Zr_0.4_O_3_	2.0 × 10^−4^ S·cm^−1^ (27 °C)	[80]
	Li_3/8_Sr_7/16_Hf_1/4_Ta_3/4_O_3_	3.8 × 10^−4^ S·cm^−1^ (RT)	[81]
	Li_3/8_Sr_7/16_Hf_1/4_Nb_3/4_O_3_	2.0 × 10^−5^ S·cm^−1^ (RT)	[82]
	Li_0.34_La_0.56_TiO_3_(film)	2.0 × 10^−5^ S·cm^−1^ (RT)	[87]
	Li_0.38_Sr_0.44_Hf_0.3_O_2.95_F_0.05_	4.8 × 10^−4^ S·cm^−1^ (25 °C)	[93]
NASICON	LiZr_2_(PO_4_)_3_	7 × 10^−4^ S·cm^−1^ (300 °C)	[99]
	Li_2.5_Sr_0.75_Zr_1.25_(PO_4_)_3_	0.178S·cm^−1^ (550 °C)	[101]
	LiTi_2_(PO_4_)_3_	2 × 10^−6^ S·cm^−1^ (25 °C)	[104]
	Li_1.3_M_0.3_Ti_1.7_(PO_4_)_3_ M = Al or Sc	7 × 10^−4^ S·cm^−1^ (25 °C)	[104]
	Li_1+x_Al_x_Ti_2-x_(PO_4_)_3_	1.3 × 10^−3^ S·cm^−1^ (RT)	[106]
	Li_1.3_Al_0.3_Ti_1.7_(PO_4_)_3_	3 × 10^−4^ S·cm^−1^ (RT)	[107]
	Li_1.4_Al_0.4_Ti_1.6_(PO_4_)_3_	1.12 × 10^−3^ S·cm^−1^ (25 °C)	[108]
	Li_1.5_Al_0.5_Ge_1.5_(PO_4_)_3_	4 × 10^−4^ S·cm^−1^ (RT)	[109]
	Li_1.5_Al_0.5_Ge_1.5_(PO_4_)_3_	4.15 × 10^−4^ S·cm^−1^ (RT)	[110]
	Li_1.5_Al_0.5_Ge_1.5_(PO_4_)_3_	3.29 × 10^−4^ S cm^−1^ (RT)	[112]
	LAGP-0.05LiO_2_	7.25 × 10^−4^ S·cm^−1^ (RT)	[113]
	Li_1.5_Al_0.33_Sc_0.17_Ge_1.5_(PO_4_)_3_	5.826 × 10^−3^ S·cm^−1^ (bulk, RT)	[114]
	Li_1.2_Al_0.2_Zr_0.1_Ti_1.7_(PO_4_)_3_	4.07 × 10^−4^ S·cm^−1^ (40 °C)	[119]
	Li_1_Zr_0.3_Ti_1.7_(PO_4_)_3_	1.84 × 10^−4^ S·cm^−1^ (40 °C)	[119]
	Li_1.2_Al_0.2_Hf_0.1_Ti_1.7_(PO_4_)_3_	2.68 × 10^−4^ S·cm^−1^ (40 °C)	[119]
	Li_1_Hf_0.3_Ti_1.7_(PO_4_)_3_	2.69 × 10^−4^ S·cm^−1^ (40 °C)	[119]
	Li_1.4_Al_0.2_Mg_0.1_Ti_1.7_(PO_4_)_3_	1.13 × 10^−4^ S·cm^−1^ (40 °C)	[119]
	Li_1.5_Al_0.1_Mg_0.2_Ti_1.7_(PO_4_)_3_	1.00 × 10^−4^ S·cm^−1^ (40 °C)	[119]
	Li_1.4_Al_0.2_Ca_0.1_Ti_1.7_(PO_4_)_3_	2.80 × 10^−4^ S·cm^−1^ (40 °C)	[119]
	Li_1.5_Al_0.1_Ca_0.2_Ti_1.7_(PO_4_)_3_	2.10 × 10^−4^ S·cm^−1^ (40 °C)	[119]
	Li_1.4_Al_0.2_Sr_0.1_Ti_1.7_(PO_4_)_3_	1.10 × 10^−4^ S·cm^−1^ (40 °C)	[119]
	Li_1.5_Al_0.1_Sr_0.2_Ti_1.7_(PO_4_)_3_	8.10 × 10^−5^ S·cm^−1^ (40 °C)	[119]
LISICON	Li_14_Zn(GeO_4_)_4_	0.125 S·cm^−1^ (300 °C)	[130]
	Li_14_Zn(GeO_4_)_4_	~10^−6^ S·cm^−1^ (RT)	[131]
	Li_3.6_Ge_0.6_V_0.4_O_4_	4 × 10^−5^ S·cm^−1^ (18 °C)	[133]
	Li_3.53_(Ge_0.75_P_0.25_)_0.7_V_0.3_O_4_	5.1 × 10^−5^ S·cm^−1^ (25 °C)	[135]
	Li_10.42_Si_1.5_P_1.5_C_l0.08_O_11.92_	1.03 × 10^−5^ S·cm^−1^ (27 °C)	[136]
	Li_10.42_Ge_1.5_P_1.5_C_l0.08_O_11.92_	3.7 × 10^−5^ S·cm^−1^ (27 °C)	[136]
Thio-LISICON	Li_2.2_Zn_0.1_Zr_0.9_S_3_	1.2 × 10^−4^ S·cm^−1^ (30 °C)	[140]
	Li_3.325_P_0.935_S_4_	1.5 × 10^−4^ S·cm^−1^ (27 °C)	[141]
	Li_3.4_Si_0.4_P_0.6_S_4_	6.4 × 10^−4^ S·cm^−1^ (RT)	[142]
	β-Li_3_PS_4_(nanoporous)	1.6 × 10^−4^ S·cm^−1^ (25 °C)	[145]
	Li_10_GeP_2_S_12_	1.2 × 10^−2^ S·cm^−1^ (RT)	[146]
	Li_3.25_Ge_0.25_P_0.75_S_4_	1.82 × 10^−4^ S·cm^−1^ (RT)	[152]
	Li_3.8_Sb_0.2_Sn_0.8_S_4_	3.5 × 10^−4^ S·cm^−1^ (RT)	[168]
	Li_10_Ge(P_1–x_Sb_x_)_2_S_12_	12.1–15.7 × 10^−3^ S·cm^−1^ (RT)	[169]
	Li_9+δ_P_3+δ_′S_11.1_O_0.9_	1.5 × 10^−3^ S·cm^−1^ (RT)	[171]
	Li_10_Sn_0.95_P_2_S_11.4_O_0.5_	3.96 × 10^−3^ S·cm^−1^ (RT)	[172]
	Li_10_SnP_1.84_Sb_0.16_S_11.6_O_0.4_	2.58 × 10^−3^ S·cm^−1^ (RT)	[173]

**Table 2 materials-16-02510-t002:** Summary of performance of ASSLB assembled with SSEs.

Electrolyte	Strategy	Cell Composition	Performance	Charge/Discharge Voltage	Ref.
LLZTO	Interfacial engineering	Li|AlN-LLZTO-AlN|Li (30 °C)	3600 h at 0.01 mA·cm^−2^.		[50]
		Li|AlN-LLZTO|LFP (30 °C)	Initial discharge capacity of 131.1 mAh·g^−1^ at 0.1 C and 122.6 mAh·g^−1^ after 200 cycles.	~3.55 V/~3.3 V	
LLZO	Interfacial engineering	Li|Janus electrolyte|Li (RT)	About 300 h at 0.2 mA·cm^−2^.		[52]
		Li|Janus electrolyte|LFP (RT)	Discharge capacity of 140 mAh·g^−1^ at 0.1 C, 128 mAh·g^−1^ at 0.2 C and negligible decay after 100 cycles.	~3.5 V/~3.3 V	
Ga-LLZO-SiO_2_	Engineering defect chemistry	Li|Ga-LLZO-SiO_2_(1 wt%)|Li (30 °C)	∼500 h at 0.2 mA·cm^−2^ and then ∼1000 h at 0.3 mA·cm^−2^		[54]
		Li|Ga-LLZO-SiO_2_(1 wt%)|LFP (30 °C)	Initial discharge capacity of 155 mAh·g^−1^ and ∼99% capacity retention after 20 cycles.	~3.5 V/~3.3 V	
Ga-LLZO	Grain refinement	Li|Ga-LLZO|Li (27 °C)	600 h at 0.4 mA·cm^−2^.		[35]
		Li|Ga-LLZO|LFP (27 °C)	Capacity of 150 mAh·g^−1^ and negligible decay after 50 cycles	~3.5 V/~3.3 V	
LLZTO	Interfacial engineering	Li|GPE@LLZO|Li (30 °C)	400 h at 0.2 mA·cm^−2^.		[58]
		Li|GPE@LLZO|LCO (30 °C)	Initial discharge capacity of 126.0 mAh·g^−1^ at 0.5 C, 104.1 mAh·g^−1^ after 100 cycles.	~4 V/~3.8 V	
LLZTO	Interfacial engineering	Li|Zn–Cu–LLZTO–Zn–Cu|Li (28 °C)	Over 450 h at 0.8 mA·cm^−2^ and a critical current density of 2.8 mA·cm^−2^.		[59]
		Li|Zn–Cu–LLZTO–Zn–Cu|LFP (28 °C)	Initial charge capacity of 146 mAh·g^−1^, 130.8 mAh·g^−1^ after 50 cycles.	~3.6 V/~3.25 V	
LLTO	Interfacial engineering	Li|PVDF:LLTO|Li (60 °C)	Fail after cycling at 0.1 mA cm^−2^ for 25 h.		[89]
		Li|PVDF:LLTO@PDA|Li (60 °C)	Over 800 h at 0.1 mA·cm^−2^.		
		Li|PVDF:LLTO@PDA|NCM622 (60 °C)	Initial charge capacity of 158.2 mAh·g^−1^ and capacity retention of 83% after 100 cycles at 0.1 C.	3.6~4.2 V/4.2~3.6 V	
LSTZ	Interfacial engineering	Li|PEO/LSTZ|Li (45 °C)	Over 700 h at 0.1 mA·cm^−2^.		[90]
		Li|PEO/LSTZ|LFP (45 °C)	A stabilized capacity of 136 mAh·g^−1^ and capacity remains at 123 mAh·g^−1^ after 350 cycles.	~3.6 V/~3.2 V	
LATP	Interfacial engineering	Li|(PAA/PEO)_30_|LATP|(PAA/PEO)_30_|Li	600 h at 0.1 mA·cm^−2^.		[121]
		Li|(PAA/PEO)_30_|LATP|(PAA/PEO)_30_|LFP	Initial discharge capacity of 115 mAh·g^−1^ at 0.1 C and 102 mAh·g^−1^ after 20 cycles.	~3.5 V/~2.6 V	
LATP	Interfacial engineering	Li|LATP@Al_2_O_3_|Li (RT)	600 h with small voltage hysteresis.		[122]
LATP	Interfacial engineering	Li|MoS_2_-LATP- MoS_2_|Li (60 °C)	More than 300 h at 0.05 mA·cm^−2^.		[123]
LATP	Interfacial engineering	Li|Cr-LATP- Cr|Li (RT)	~850 h at 0.2 mA·cm^−2^.		[124]
LAGP	Interfacial engineering	Li|Li_2_OHBr|LAGP|Li_2_OHBr|Li (RT)	300 h at 0.05 mA·cm^−2^.		[125]
		Li|Li_2_OHBr|LAGP|Li_2_OHBr|LFP (RT)	Initial specific capacity of 119.9 mAh·g^−1^, and remains the capacity of 110.1 mAh·g^−1^ after 20th cycle and the capacity of 96.3 mAh·g^−1^ after the 40th cycle at 0.1 C.	~3.45 V/~3.35 V	
LATP	Interfacial engineering	Li|5SnO_2_@LATP|LNLO|NCM (RT)	Initial discharge capacity of 171.49 mAh·g^−1^ and 89.47 % capacity retention after 100 cycles at 0.1C.	3.6~4.3 V/4.3~3.3 V	[129]
LGPS	Interfacial engineering	Li_0.8_Al|LGPS|Li_0.8_Al	2500 h at 0.5 mA·cm^−2^		[156]
LGPS	Interfacial engineering	Li|Nanocomposites-LGPS- Nanocomposites|Li	Over 1700 h with a Li deposition amount of 0.2 mAh·g^−1^.		[157]
LGPS	Interfacial engineering	Li-LiH_2_PO_4_|LGPS|LiH_2_PO_4_-Li (25 °C)	Over 950 h at 0.1 mA·cm^−2^.		[158]
		Li-LiH_2_PO_4_|LGPS|LCO (25 °C)	113.7 mAh·g^−1^ for the 500th cycle at 0.1 C with a retention of 86.7%.	~3.9 V/~3.9 V	
Li_3.8_Sb_0.2_Sn_0.8_S_4_		LTO| Li_3.8_Sb_0.2_Sn_0.8_S_4_|LCO (RT)	Initial discharge capacity of up to 125 mAh·g^−1^, and gradually decreased to 105 mAh·g^−1^ (84% of the first discharge capacity) after 10 cycles.		[168]
Li_10_Ge(P_0.925_Sb_0.075_)_2_S_12_		In|Li_10_Ge(P_0.925_Sb_0.075_)_2_S_12_|LNO@LCO (25 °C)	Initial discharge capacity of 128 mAh·g^−1^ at 0.1 C and 108 mAh·g^−1^ remained for over 50 cycles.	3.3~3.6 V/3.6~3.3 V	[169]
Li_10_Sn_0.95_P_2_S_11.4_O_0.5_		Li-In|Li_10_Sn_0.95_P_2_S_11.4_O_0.5_|LNO@LCO (RT)	Specific discharge capacity of 133 mAh·g^−1^ in the first cycle.	3.1~3.6 V/3.5~2.9 V	[172]
Li_10_SnP_1.84_Sb_0.16_S_11.6_O_0.4_		Li-In|Li_10_SnP_1.84_Sb_0.16_S_11.6_O_0.4_|LNO@LCO (25 °C)	Initial discharge capacity of 96 mAh·g^−1^ at 0.5 C, and maintains 85% capacity retention after 200 cycles.	3.3~3.6 V/3.6~3.3 V	[173]
LGPS	Interfacial engineering	Li|40s air-exposed Li_10_GeP_2_S_12_|Li (25 °C)	1000 h with small polarization voltage of 26 mV at 0.1 mA·cm^−2^.		[174]
		Li|40s air-exposed Li_10_GeP_2_S_12_|LCO (25 °C)	100 cycles with capacity retention of 80%, and discharge capacity of 113, 87, 66, 46 mAh·g^−1^ at 0.1, 0.2, 0.5 and 1 C, respectively.	~3.9 V/~3.8 V	

## Data Availability

Not applicable.

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
