# Peer review of "Review of the Developments and Difficulties in Inorganic Solid-State Electrolytes"

_materials, 2023, doi:10.3390/ma16062510_

Round 1

Reviewer 1 Report

   1.  Many papers on same topic have already been published. Please justify how your paper is different from the others.

2 2.     Information provided at several places in the manuscript is not correct. Such as, as said in line 51the “Inorganic Li ion conductors have the advantage of high structural stability” is not correct as the tetragonal Li7La3Zr2O12 Solid state electrolyte or Ga-doped Li7La3Zr2O12 solid state electrolyte has poor structural stability at room temperature. Also, in line 184 as said the “LLZO with high density could prevent dendrite growth and penetration” has been proved wrong in several publications. Such problem has been identified elsewhere in the manuscript.

3 3. Mention numeric value of the operating temperature range, electrochemical stability window, ionic conductivity, activation energy, etc. with reference in the manuscript like these values has been mentioned rightly while describing thio-LISICON-type Li10GeP2S12 (LGPS). Also, the authors should mention the density of conventional Li-ion battery and all solid-state batteries with Li anode with reference.

44.  Authors should speak more about widely explored Ga-doped LLZO in section 2.1.1.

 5.  Recent Reference “Inorganic Solid-State Electrolytes: Insights on Current and Future Scope” by Atul k Mishra et al. should be cited.

Author Response

Reviewer 1. Comments and Suggestions for Authors

  1.    Many papers on same topic have already been published. Please justify how your paper is different from the others.

Author reply: We would like to thank you for reviewing our paper, we appreciate your insightful comments on our research. Firstly, this article provides a comprehensive explanation of four typical inorganic SSEs. From the historical development, to the structure and ion migration mechanism, as well as the optimization of various electrolytes by doping is discussed. Different from other articles, in this article, for each type of SSE, the stability toward anode/cathodes and air stability are analyzed respectively, and the recent solutions are introduced. Afterward, the properties of the SSEs are summarized, and the advantages and disadvantages of the four inorganic SSEs are pointed out. The performances of the ASSLIBs mentioned in the article are summarized and the performance difference between commercial Li-ion batteries and laboratory ASSLIBs is compared, which is seldom mentioned in other articles. In the end, an outlook for SSEs from a commercialization perspective is proposed.

  1. Information provided at several places in the manuscript is not correct. Such as, as said in line 51the “Inorganic Li ion conductors have the advantage of high structural stability” is not correct as the tetragonal Li7La3Zr2O12 Solid state electrolyte or Ga-doped Li7La3Zr2O12 solid state electrolyte has poor structural stability at room temperature. Also, in line 184 as said the “LLZO with high density could prevent dendrite growth and penetration” has been proved wrong in several publications. Such problem has been identified elsewhere in the manuscript.

Author reply: Thank you for your advice. Efforts were made to check the article. Some wrong information has been deleted, and some information has been optimized so that make it more accurate.

  1. Mention numeric value of the operating temperature range, electrochemical stability window, ionic conductivity, activation energy, etc. with reference in the manuscript like these values has been mentioned rightly while describing thio-LISICON-type Li10GeP2S12 (LGPS). Also, the authors should mention the density of conventional Li-ion battery and all solid-state batteries with Li anode with reference.

Author reply: Thank you for your advice. The conductivity, activation energy, electrochemical stability window and other properties of some electrolytes are supplemented in the manuscript. The capacity density of commercial lithium-ion batteries was investigated and mentioned in the introduction and summary section. In the summary part, the performance of commercial LIBs and ASSLIBs in the laboratory are compared to show the advantages and shortages of ASSLIB.

  1. Authors should speak more about widely explored Ga-doped LLZO in section 2.1.1.

Author reply: Thank you for your advice. The development, performance and optimization of Ga-LLZO are added in 2.1.1. In addition, the mechanism of low stability between Ga-LLZO and Li anode, and modification measures are added in 2.1.2.

  1. Recent Reference “Inorganic Solid-State Electrolytes: Insights on Current and Future Scope” by Atul k Mishra et al. should be cited.

Author reply: Thank you for your advice. The article is referred at the Ref. [13]

Reviewer 2 Report

Dear Authors! Thank you for your manuscript, submitted to "Materials", which is devoted to a comprehensive review of the all-solid electrolytes for Li-ion batteries, their structure, Li-ion transport mechanism, and encountered difficulties, including interfaces and and air stability.

I believe, that the submitted review article, due to its simple and clear presentation, will be of interest to the readers and useful for scientists and students, who want to get acquainted with the above-mentioned problems. At the same time, I think that the manuscript should be subjected to extensive corrections prior to publication in order to improve it. I  would like to propose to draw attention to some crucial moments:

1. It is no secret, that there are many published reviews, dedicated to electrolytes for Li-ion batteries. According to Scopus, 151 reviews have been published in the period 2020-2023. And some of them have a high number of citations (e.g., review on garnet-type electrolytes, 2020, 379 citations, DOI:10.1021/acs.chemrev.9b00427; review on interfaces stability - 2020, 413, 10.1038/s41578-019-0157-5, and so on)

Therefore, it is necessary to emphasise the novelty of the submitted work. 

2. It is also necessary to update the relevant references by adding the 2020-2023 references, because of: Paragraph 2.1 contains 26 references of 2020-2022 period out of 48; p. 2.2 - only 5 out of 20; p. 2.3 - 9 out of 29; p. 2.4 - only 7 out of 37.

3. There is a lack of correlation analysis of data for different types of electrolytes observed. For example, there are may be tables with conductivity, infographics with conductivity, electrochemical stability window, time stability, interfacial resistance, etc.  

Author Response

Reviewer 2. Dear Authors! Thank you for your manuscript, submitted to "Materials", which is devoted to a comprehensive review of the all-solid electrolytes for Li-ion batteries, their structure, Li-ion transport mechanism, and encountered difficulties, including interfaces and and air stability.

I believe, that the submitted review article, due to its simple and clear presentation, will be of interest to the readers and useful for scientists and students, who want to get acquainted with the above-mentioned problems. At the same time, I think that the manuscript should be subjected to extensive corrections prior to publication in order to improve it. I  would like to propose to draw attention to some crucial moments:

  1. It is no secret, that there are many published reviews, dedicated to electrolytes for Li-ion batteries. According to Scopus, 151 reviews have been published in the period 2020-2023. And some of them have a high number of citations (e.g., review on garnet-type electrolytes, 2020, 379 citations, DOI:10.1021/acs.chemrev.9b00427; review on interfaces stability - 2020, 413, 10.1038/s41578-019-0157-5, and so on)

Therefore, it is necessary to emphasise the novelty of the submitted work. 

Author reply: We would like to thank you for reviewing our paper, we appreciate your insightful comments on our research. Here are some novelties:

(1) In addition to discussing the basic development, structure, and Li-ion transport mechanism of solid electrolytes, in this article, for each type of SSE, the stability toward anode/cathodes and air stability are analyzed respectively, and the recent modifications and solutions are introduced.

(2) In addition to the properties summary of the SSEs, the performance of the battery mentioned in the article is summarized. The advantages and disadvantages of four typical inorganic electrolytes are pointed out, and the ASSLIBs are compared with the commercialized LIB to show the advantages and shortages of ASSLIBs.

  1. It is also necessary to update the relevant references by adding the 2020-2023 references, because of: Paragraph 2.1 contains 26 references of 2020-2022 period out of 48; p. 2.2 - only 5 out of 20; p. 2.3 - 9 out of 29; p. 2.4 - only 7 out of 37.

Author reply:  Thank you for your advice. More articles in 2020-2023 are read and introduced in the article. Now, paragraph 2.1 contains 30 references of 2020-2022 period out of 55; p. 2.2 - 8 out of 25; p. 2.3 - 16 out of 36; p. 2.4 - only 15 out of 45. As the historical process and development of the electrolytes are introduced, some references to older articles are unavoidable.

  1. There is a lack of correlation analysis of data for different types of electrolytes observed. For example, there are may be tables with conductivity, infographics with conductivity, electrochemical stability window, time stability, interfacial resistance, etc.  

Author reply:  Thank you for your advice. The 3rd section summary is added in this review. The properties of various SSEs and the performances of ASSLIBs mentioned in the article are summarized in two tables. The different advantages and disadvantages of the typical inorganic SSEs are pointed out.

Reviewer 3 Report

Authors discussed some of the recent developments of the solid electrolytes, which is worth considering after its revision  with listed comments.

Similar conceptual summarized reviews published recently. However, in this current review, the latest works are discussed, which brings an advantage to interested researchers. But the following observations and comments should be addressed to improve its quality further.

  Review comments:

1. Abstract should be improved.

2. Several typo errors were found in the overall manuscript, authors need to check carefully to revise and verify.

3. Through revision is required for the Introduction section and can be improved.

4.  Authors can elaborate/extend the discussion in detail for section 2.4.5. ( in most of the published journals this topic is not explained distinctly so this is can be considered ).

Section 2.4.5 is dedicated to Air Stability which is really interesting and needs more understanding/analysis to accelerate its progress. So this section should be discussed along with its perspectives. Authors mentioned "Additionally, difficulties such as interface  issues and air stability as well as possible solutions are also discussed."  These two points are real time issues to discuss in detail . Section 2.4.5 is dedicated to Air Stability but cannot find much details or their mechanisms/analysis.

Section 2.4.5 need to explain with figures (Mechanisms/analysis). Summarized table will help readers  for a quick reference.

Author Response

Reviewer 3:

Authors discussed some of the recent developments of the solid electrolytes, which is worth considering after its revision  with listed comments.

Similar conceptual summarized reviews published recently. However, in this current review, the latest works are discussed, which brings an advantage to interested researchers. But the following observations and comments should be addressed to improve its quality further.

  Review comments:

  1. Abstract should be improved.

Author reply:  We would like to thank you for reviewing our paper, we appreciate your insightful comments on our research. Abstract has been improved.

  1. Several typo errors were found in the overall manuscript, authors need to check carefully to revise and verify.

Author reply:  Thank you for your advice. Typo and grammar are checked and corrected in the article.

  1. Through revision is required for the Introduction section and can be improved.

Author reply:  Thank you for your advice. Introduction has been improved.

  1. Authors can elaborate/extend the discussion in detail for section 2.4.5. ( in most of the published journals this topic is not explained distinctly so this is can be considered ).

Section 2.4.5 is dedicated to Air Stability which is really interesting and needs more understanding/analysis to accelerate its progress. So this section should be discussed along with its perspectives. Authors mentioned "Additionally, difficulties such as interface  issues and air stability as well as possible solutions are also discussed."  These two points are real time issues to discuss in detail . Section 2.4.5 is dedicated to Air Stability but cannot find much details or their mechanisms/analysis.

Section 2.4.5 need to explain with figures (Mechanisms/analysis). Summarized table will help readers for a quick reference.

Author reply:  Thank you for your advice. section 2.4.5 which is mainly about the air stability of thio-LISICON has been extended. The chemical reason and mechanism are discussed in detail and a figure is drawn to explain the mechanism. As the most efficient method to improve the stability of thio-LISICON SSEs, doping strategies based on hard and soft acids and bases (HSAB) theory is introduced.

The 3rd section summary is added in this review. The properties of various SSEs and the performances of ASSLIBs mentioned in the article are summarized with references.

Round 2

Reviewer 2 Report

Dear Authors! Thank you for your great attention to my comments. Your manucsript has been improved, no doubt. It is the comprehensive review now, ready for publication.

Reviewer 3 Report

The authors made significant Modifications to improve the manuscript and organize it properly.